# ScoreMix: Synthetic Data Generation by Score Composition in Diffusion Models Improves Recognition

## Abstract

Synthetic data generation is increasingly used in machine learning for **training and data augmentation**. Yet, current strategies often rely on external foundation models or datasets, whose usage is restricted in many scenarios due to policy or legal constraints. We propose **ScoreMix**, a **self-contained** synthetic generation method to produce hard synthetic samples for recognition tasks by leveraging the score compositionality of diffusion models. The approach mixes class-conditioned scores along reverse diffusion trajectories, yielding domain-specific data augmentation without external resources. We systematically study class-selection strategies and find that mixing classes distant in the discriminator's embedding space yields larger gains, providing **up to 3% additional average improvement**, compared to selection based on proximity. Interestingly, we observe that condition and embedding spaces are largely uncorrelated under standard alignment metrics, and the generator's condition space has a negligible effect on downstream performance. Across **8 public face recognition benchmarks**, ScoreMix improves accuracy by **up to 7 percentage points**, without hyperparameter search, highlighting both robustness and practicality. Our method provides a simple yet effective way to maximize discriminator performance using only the available dataset, without reliance on third-party resources. *Code and synthetic datasets are available.*

## 1 Introduction

Synthetic dataset generation has emerged as a powerful tool for training models across a wide range of domains. A central application of this paradigm is **data augmentation**, which is indispensable for training strong discriminators, particularly when labeled data is limited. However, most existing strategies depend on external resources such as large foundation models or auxiliary datasets that are often impractical due to license restrictions, privacy concerns, or mismatched domains. This raises a central question: *can we design a* ***self-contained*** *augmentation method that leverages only the available dataset to generate synthetic data and boost discriminative performance?*

This paper introduces **ScoreMix**, an augmentation strategy that exploits the *score composition phenomenon* in diffusion models (Liu et al., 2022; Bradley et al., 2025).

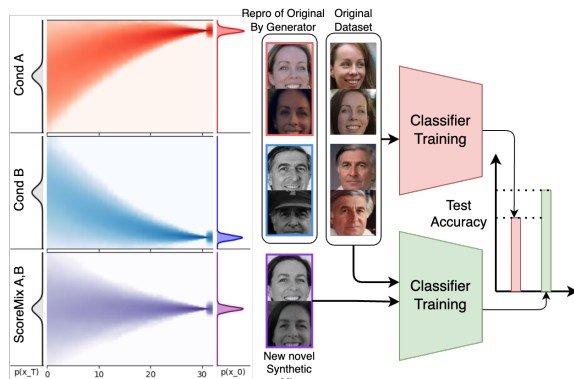

Figure 1: ScoreMix. Adding carefully generated synthetic augmentations to the original training set boosts the discriminator's performance, without relying on other sources of information (right). The first two subplots on the left show diffusion trajectories obtained under two different conditioning signals (Cond A/B). Using convex combinations of their score functions (ScoreMix A,B), we generate synthetic samples that interpolate between the two trajectories.

Rather than relying on external generators such as Stable Diffusion (Esser et al., 2024) or FLUX (Labs, 2024), or even strong pre-trained backbones like SigLIP (Tschannen et al., 2025), ScoreMix produces synthetic data with convex combinations of class-conditioned scores during reverse diffusion. Crucially, both the generator and the initial discriminator are trained from scratch on the same dataset, ensuring a fully self-contained setup. This procedure yields hard on-manifold samples that enrich the training set with no information leakage.

We summarize the goal and contributions of this paper as follows.

> • **Goal:** to develop a **self-contained** augmentation strategy—that is, one that does not rely on external datasets, commercial APIs, or third-party models—to maximize the performance of state-of-the-art discriminators solely with the available data.

- **Synthetic augmentation via score mixing.** We demonstrate that convex combinations of class-conditioned scores yield synthetic samples that consistently improve discriminator training.
- **Performance improvement.** Across eight face recognition benchmarks, ScoreMix improves accuracy by up to **7%** without any hyperparameter search. It not only surpasses training on the original dataset but also outperforms architectural scaling (e.g., IR101 vs. IR50) or higher training iterations, underscoring the practical advantages of self-contained synthetic augmentation.
- **Class selection analysis.** We show that mixing classes that are *distant in the discriminator's embedding space* produces the largest gains, while proximity in the generator's condition space has little effect.
- **Geometry and alignment.** We empirically reveal that the generator's condition space and the discriminator's embedding space are only weakly correlated under standard metrics, highlighting possible causes of why condition-based selection underperforms.
- **Theoretical robustness.** We establish order-preserving probability guarantees between the generator's condition space and the discriminator's embedding space under common alignment metrics such as CKA, showing that class selection remains effective across different backbones.

## 2 RELATED WORK

Synthetic data generation is widely explored as an alternative to large-scale data collection. Early augmentation strategies are based on GANs (Frid-Adar et al., 2018) but do not scale well with the number of classes. Recent approaches use diffusion models, e.g. fine-tuning on ImageNet (Azizi et al., 2023), instance-level redraws (Kupyn & Rupprecht, 2024), and 3DMM-based rendering (Wood et al., 2021; Blanz & Vetter, 1999). These are effective but depend on external pretrained models or datasets. Face recognition (FR) is an important application of synthetic augmentation, with methods such as SynFace (Qiu et al., 2021), StyleGAN-based latent modeling (Rahimi et al., 2023), dual-condition diffusion (DCFace (Kim et al., 2023)), StyleGAN2-ADA for bias mitigation (Sevastopolskiy et al., 2023), attribute-conditioned diffusion (ID3 (Xu et al., 2024)), and 3D rendering pipelines like DigiFace1M and RealDigiFace (Bae et al., 2023; Rahimi et al., 2024) and CLIP-guided sampling (VariFace (Yeung et al., 2024)). FR is attractive because collecting diverse face datasets is difficult. Benchmarks such as LFW (Huang et al., 2008), IJB-B/C (Whitelam et al., 2017), and AgeDB (Moschoglou et al., 2017) provide more reliable testing protocols than noisier ImageNet settings. Recently, Rahimi et al. (2025) introduced a self-contained strategy to produce challenging samples (AugGen). They train a diffusion generator on a target FR dataset and mix labels in the generator's condition space. This relies on heuristics and a costly parameter search. Our work builds on this work while addressing its limitations, by leveraging score composition in diffusion models and aligning class selection with the geometry of the discriminator's embedding space.

## 3 PROPOSED METHOD FOR GENERATING AUGMENTATIONS

We first formally define the notion of a discriminator and a generator trained using the same dataset.

**Discriminator.** Assume a dataset $\mathbf{D}_{\text{orig}} = \{(\mathbf{X}_i, y_i)\}_{i=0}^{k-1}$, where each $\mathbf{X}_i \in \mathbb{R}^{H \times W \times 3}$ and $y_i \in \{0, \dots, l-1\}$ ($l < k$). The goal is to learn a discriminative model $f_{\theta_{\text{dis}}} : \mathbf{X} \to \mathbf{y}$ that estimates $p(\mathbf{y}|\mathbf{X})$ (e.g., on ImageNet (Russakovsky et al., 2015) or CASIA-WebFace (Yi et al., 2014)). Typically, similar images have closer features under a distance $\text{dist}_{\text{emb}}$ (e.g., cosine distance). We train $f_{\theta_{\text{dis}}}$ via empirical risk minimization:

$$\theta_{\text{dis}}^* = \underset{\theta_{\text{dis}} \in \Theta_{\text{dis}}}{\arg\min} \, \mathbb{E}_{(\mathbf{X},y) \sim \mathbf{D}_{\text{orig}}} \big[ \mathcal{L}_{\text{dis}}(f_{\theta_{\text{dis}}}(\mathbf{X}), \mathbf{y}) \big], \tag{1}$$

where $\mathcal{L}_{\mathrm{dis}}$ is typically cross-entropy, and $\mathrm{h}_{\mathrm{dis}}$ manifests all the hyperparameters (e.g., learning rates).

**Generative model.** Generative models seek to learn the data distribution, enabling the generation of new samples. We use diffusion models (Song et al., 2020; Anderson, 1982), which progressively add noise to data and train a Denoiser S. Following (Karras et al., 2022; 2024b), S is learned in two stages. First, for a given noise level $\sigma$, we add noise $\mathbf{N}$ to $E_{\mathrm{pre}}(\mathbf{X})$ (or $\mathbf{X}$ directly in pixel-based diffusion) and remove it via:

$$\mathcal{L}(\mathrm{S}_{\theta_{den}}; \sigma) = \mathbb{E}_{(\mathbf{X}, y) \sim \mathrm{D}^{\mathrm{orig}}, \mathbf{N} \sim \mathcal{N}(\mathbf{0}, \sigma\mathbf{I})} \left[ \| \mathrm{S}_{\theta_{den}}(E_{\mathrm{pre}}(\mathbf{X}) + \mathbf{N}; \mathrm{c}(y), \sigma) - \mathbf{X} \|_2^2 \right], \qquad (2)$$

where $\mathrm{c}(y)$ denotes the class condition, and $E_{\mathrm{pre}}(\cdot)$ and $D_{\mathrm{pre}}(\cdot)$ pre-processing and post-processing functions in terms of Encoder and Decoder (*e.g.*, they can be magnitude normalization or VAE-based compression). In the second stage, we sample different noise levels and minimize:

$$\theta_{den}^* = \underset{\theta_{den} \in \Theta_{den}}{\arg\min} \; \mathbb{E}_{\sigma \sim \mathcal{N}(\mu, \sigma^2)} \left[ \lambda_\sigma \, \mathcal{L}(\mathrm{S}_{\theta_{den}}; \sigma) \right], \qquad (3)$$

where $\lambda_\sigma$ weights each noise scale. Here $\boldsymbol{c}$ amongst the time embedding is learned. For simplicity, we omit the *den* and *dis* subscripts used to distinguish the parameters of the Denoiser and Discriminator, respectively. Instead, we use $\theta$ to denote parameters in general, with the specific meaning inferred from context.

**Conditional score estimation in diffusion models.** The predicted noise depicted in the previous section is proportional to the score function $\nabla_{\mathbf{X}_t} \log p_t(\mathbf{X}_t | c)$ (Song et al., 2020; Karras et al., 2024b). Given two distinct conditions, $c_A$ and $c_B$, we can obtain their respective conditional score predictions:

$$\mathbf{S}_A(\mathbf{X}_t, t) = \mathrm{S}_\theta(\mathbf{X}_t, t, c_A) \qquad \mathbf{S}_B(\mathbf{X}_t, t) = \mathrm{S}_\theta(\mathbf{X}_t, t, c_B) \qquad (4)$$

Our work aims to generate novel synthetic data augmentations by composing information from two or more distinct conditional distributions learned by a diffusion model. We achieve this by linearly combining their respective score estimates during the reverse diffusion process.

## 3.1 SYNTHETIC AUGMENTATION VIA CONVEX SCORE MIXING

To generate synthetic samples that interpolate or combine aspects of both $c_A$ and $c_B$, we propose a mixed score $\mathbf{S}_{\mathrm{mix}}$:

$$\mathbf{S}_{\mathrm{mix}}(\mathbf{X}_t, t) = \alpha \cdot \mathbf{S}_A(\mathbf{X}_t, t) + \beta \cdot \mathbf{S}_B(\mathbf{X}_t, t) \qquad (5)$$

This mixed score $\mathbf{S}_{\mathrm{mix}}$ is then used to guide the denoising step in a standard reverse diffusion sampler (e.g., DDIM (Song et al., 2020) or a second-order solver as in (Karras et al., 2024b)). Prior works have explored linear combinations of scores for compositional generation, often aiming to satisfy product-of-experts-like objectives or achieve disentangled concept manipulation (Liu et al., 2022; Bradley et al., 2025). These works typically focus on composing disparate concepts (e.g., "object" + "style") or attributes.

In our work, we adapt this principle specifically for generating *nuanced synthetic augmentations* by mixing related conditional distributions. We hypothesize that for this application, maintaining the overall magnitude and directional integrity of the score is paramount for generating plausible, on-manifold samples. To the best of our knowledge, this is the first work to systematically investigate and leverage this form of multi-conditional score mixing specifically for the task of generating synthetic data augmentations that lie "between" two defined conditional states, effectively generating hard samples for the discriminator to further boost its discriminative and increase the chance of capturing any missed information from the initial training on of the discriminator.

We empirically find that the most plausible and high-fidelity synthetic augmentations are generated when the mixing coefficients $\alpha$ and $\beta$ form a convex combination ($\alpha + \beta = 1$). The theoretical rationale for this observation is rooted in several properties of score-based models.

- **Preservation of expected score magnitude.** Diffusion models, particularly those with stabilized training dynamics like EDM2 (Karras et al., 2024b), are trained such that the predicted noise (and thus the score) has an expected magnitude appropriate for the current noise level $t$. A convex combination $\lambda \mathbf{S}_A + (1 - \lambda) \mathbf{S}_B$ inherently averages the directional vectors while being more likely to preserve an overall magnitude consistent with what the model expects. If $\alpha + \beta \gg 1$, the resulting score magnitude might become excessively large, akin to an extreme guidance scale in classifier-free guidance (Ho & Salimans, 2021), potentially pushing samples off the manifold. Conversely, if $\alpha + \beta \ll 1$, the score magnitude might be too small, leading to under-denoising.

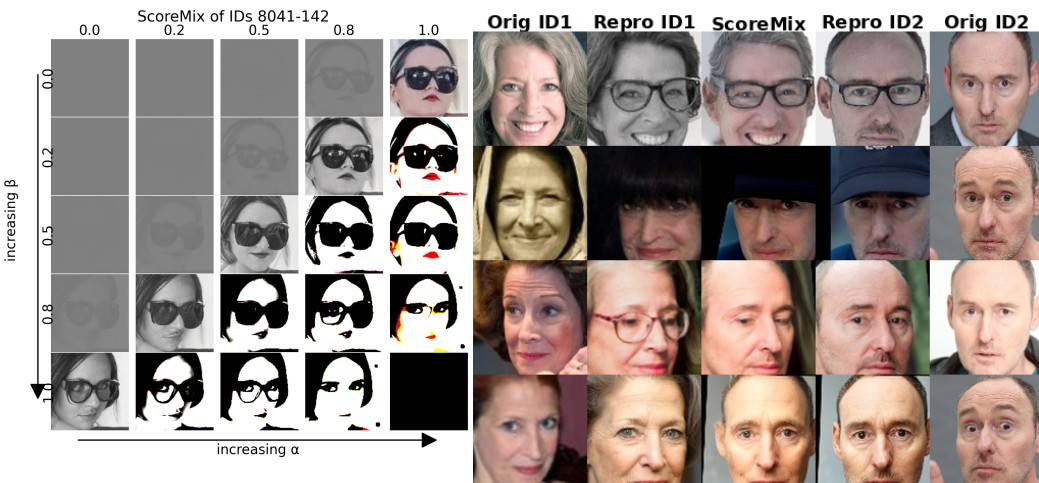

Figure 2: Effect of mixing scores in SCOREMIX. Each cell shows the image produced for one pair of inputs while sweeping $\alpha$ (horizontal, left→right) and $\beta$ (vertical, top→bottom). Randomness is fixed across images.

Figure 3: Qualitative comparison of ScoreMix augmentation. Rows show *Orig ID1*, *Repro ID1*, *ScoreMix* (Eq. 5, AutoGuidance=1.3), *Repro ID2*, and *Orig ID2*. The center column provides augmented samples whose subtle deviations from original ones improve discriminator performance.

- **Interpolation on the Data Manifold.** The score vectors $\mathbf{S}_A$ and $\mathbf{S}_B$ point towards regions of the data manifold consistent with $c_A$ and $c_B$, respectively. A convex combination provides a principled way to interpolate the "denoising force" along paths on this learned manifold. Non-convex combinations could result in update directions that lead to low-density regions or out-of-distribution samples. This will be highlighted empirically in the next section.
- **Factorized Conditionals and Projective Composition.** Recent theoretical work (Bradley et al., 2025) suggests that linear score combinations can provably achieve a desired "projective composition" under certain conditions, such as when the underlying distributions exhibit a factorized conditional structure or can be mapped to such a structure in a feature space. While our conditions $c_A$ and $c_B$ may not always strictly satisfy these assumptions (e.g., if they represent entangled attributes), a convex mixing provides the most stable approximation for interpolating between them by maintaining a consistent update scale. Interestingly, as we will highlight in the next section that even though we target identity mixing, the generated samples provably improve the performance of the discriminator.

The architectural advancements in models like EDM2 (Karras et al., 2024b), which focus on preserving activation and weight magnitudes, further bolster the argument for convex combinations. If individual conditional scores are already well-calibrated by the model architecture, their convex mix is one of the plausible ways to fuse their guidance without introducing extraneous magnitude distortions.

In Figure 2, the effect of different values of $\alpha$ and $\beta$ is depicted. Numeric tick labels give the exact values in steps of $0.2$. Here, the class conditional generator is trained using face images in which each class is a unique identity. Arrows beneath and at the side of the grid highlight the directions of increasing influence from each source. The extreme corner corresponds to the unmixed original scores $((\alpha, \beta) = (0, 0)$ at the top-left and equivalently mixed $(1, 1)$ at the bottom-right, while the descending diagonal where $\alpha + \beta = 1$ illustrates the complementary trade-off between the two sources; off-diagonal cells reveal The visual behaviour when the weights do *not* sum to 1, which empirically reflects our previous discussion. See Appendix L for more samples.

## 3.2 SAMPLING PROCEDURE

For generating samples, we employ the deterministic second-order sampler detailed in (Karras et al., 2024b; 2022). At each step $t$, the mixed score $\mathbf{S}_{\text{mix}}(\mathbf{X}_t, t)$ from Equation 5 is used in place of the single conditional score to compute the update $\Delta\mathbf{X}_t$. The specific mixing parameter $\lambda$ (where $\alpha = 1 - \lambda, \beta = \lambda$) could be varied to generate a spectrum of synthetic augmentations. For simplicity and intuition, we set the $\lambda = 0.5$. Given the conditions $c_A$ and $c_B$, the detailed algorithmic procedure

for mixing the conditions to generate a plausible mixed image is presented in Algorithm 1. We are also applying autoguidance (Karras et al., 2024a) for sampling, with a model trained with fewer iterations. Some examples of the ScoreMix samples are depicted in the middle column of Figure 3. See Appendix L for more samples.

---

**Algorithm 1** Sampling with Convex Conditional Score Mixing

**Require:** Denoising network $S_\theta(\mathbf{X}_t, t, c)$; conditions $c_A$, $c_B$; weights $\alpha=0.5$, $\beta=0.5$; Solver steps $T$
1  Initialize $\mathbf{X}_t \sim \mathcal{N}(\mathbf{0}, \sigma_T^2 \mathbf{I})$      ▷ Sample initial noise
2  **for** $t = T$ down to 1 **do**
3     $\mathbf{S}_A \leftarrow S_\theta(\mathbf{X}_t, t, c_A)$      ▷ Predict noise for A
4     $\mathbf{S}_B \leftarrow S_\theta(\mathbf{X}_t, t, c_B)$      ▷ Predict noise for B
5     $\mathbf{S}_{\text{mix}} \leftarrow \alpha \cdot \mathbf{S}_A + \beta \cdot \mathbf{S}_B$      ▷ Convex combination
6     $\boldsymbol{x}_{t-1} \leftarrow \text{SamplerStep}(\mathbf{X}_t, t, \mathbf{S}_{\text{mix}})$ ▷ Update with mixed score
7  **end for**
**Ensure:** Final generated sample $\boldsymbol{x}_0$      ▷ Output image

---

**Algorithm 2** DISTANCECORRELATION

**Require:** $E \in \mathbb{R}^{l \times d_E}$ with $\text{dist}_{\text{emb}}$; $C \in \mathbb{R}^{l \times d_C}$ with $\text{dist}_{\text{cond}}$
**Ensure:** $\mathbf{e}$, $\mathbf{c}$
1  $\mathbf{e} \leftarrow [\,]$, $\mathbf{c} \leftarrow [\,]$      ▷ Init lists
2  **for** $i \leftarrow 1$ to $l - 1$ **do**
3     **for** $j \leftarrow i + 1$ to $l$ **do**
4       $u \leftarrow \text{dist}_{\text{emb}}(E_{:,i}, E_{:,j})$
5       $v \leftarrow \text{dist}_{\text{cond}}(C_{:,i}, C_{:,j})$
6       append $u$ to $\mathbf{e}$ & append $v$ to $\mathbf{c}$
7     **end for**
8  **end for**

---

## 4 EXPERIMENTS

We show that ScoreMix improves face recognition (FR) under limited data, a critical setting given the difficulty of collecting large facial datasets. As FR requires distinguishing between millions of identities in a structured input space, it remains one of the most challenging discriminative tasks, which utilizes SOTA discriminative models that use margin losses (Deng et al., 2019).

### 4.1 EXPERIMENTAL SETUP

**Training data.**   We use WebFace160K (Rahimi et al., 2025), a subset of WebFace4M (Zhu et al., 2021), selected for its balanced distribution of 10,000 identities with 11–24 samples each ( 160K images), matching the scale of commonly used datasets like CASIA-WebFace (Yi et al., 2014). We choise this dataset over CASIA-WebFace due to performance inconsistencies previously reported in (Rahimi et al., 2025). See the Appendix I for details.

**Discriminative model.**   We adopt a standardized baseline. This baseline employs a face recognition (FR) system consisting of an IR50 backbone, modified according to the ArcFace's implementation (Deng et al., 2019), paired with the ArcFace head (Deng et al., 2019) to incorporate margin loss. Additionally, standard augmentations for face recognition tasks are applied to all models. These augmentations include (1) photometric transformations (2) cropping, and (3) low-resolution adjustments to simulate common variations encountered in real-world scenarios. See Appendix J for details.

**Generative model.**   To train our generative model, we use a variant of the diffusion formulation (Karras et al., 2022; 2024b). For WebFace160K(Rahimi et al., 2025), the subset of WebFace4M(Yi et al., 2014), we use the pixel space variant diffusion models. Furthermore, the conditions are learned end-to-end using a diffusion objective with no explicit regularization.

### 4.2 EXPERIMENTS ON FACE RECOGNITION BENCHMARKS

**FR benchmarks.**   We evaluate our synthetic augmentation on two groups of public FR benchmarks. The first group (**Avg-H** in Table 1) contains **H**igh-quality datasets with variation in pose, lighting, and age: LFW (Huang et al., 2008), CFPFP (Sengupta et al., 2016), CPLFW (Zheng & Deng, 2018), CALFW (Zheng et al., 2017), and AgeDB (Moschoglou et al., 2017). The second group captures more realistic and challenging conditions: IJB-B/C (Maze et al., 2018; Whitelam et al., 2017) and TinyFace (Cheng et al., 2019). Evaluation is based on verification accuracy (TAR), with thresholds from cross-validation for **H**igh-quality datasets and fixed FPRs ($10^{-6}$ and $10^{-5}$) for IJB-B/C, reflecting deployment scenarios.

Table 1 also reports whether auxiliary models/datasets are used or not (**Aux**; the ideal case being N), and the training set sizes in terms of synthetic ($n^s$) and real ($n^r$) images. Following (Rahimi et al., 2025), as mentioned earlier, we adopt WebFace160K due to inconsistencies in CASIA-WebFace; results using different base datasets are separated by a double line. While ScoreMix roughly doubles the computational cost of AugGen, it consistently outperforms both AugGen and training on the

original dataset across IR50 and even surpasses the stronger IR101 backbone trained on the original dataset, indicating that augmentation can yield greater gains than architectural scaling.

> **Takeaway:** ScoreMix with $\lambda = 0.5$ consistently improves discriminator performance when trained with a single dataset for synthetic data generation, surpassing the original discriminator and outperforming larger models.

Table 1: Comparison of the $\text{FR}_{\text{syn}}$ training (upper part), $\text{FR}_{\text{real}}$ training (middle), and $\text{FR}_{\text{mix}}$ training (bottom) using CASIA-WebFace/WebFace160K, when the models are evaluated in terms of accuracy against standard FR benchmarks. **Avg-H** depicts the average accuracy of all high-quality benchmarks. Here $n^s$ and $n^r$ depict the number of Synthetic and Real Images, respectively, and Aux depicts whether the method for generating the dataset uses an auxiliary information network for generating the datasets (Y) or not (N). The † denotes network trained on IR101 if not the model trained using the IR50. The numbers under columns labeled like C/B-1e-6 indicate TAR for IJB-C/B at FPR of 1e-6. TR1 depicts the rank-1 accuracy for the TinyFace benchmark.

| Method/Data | Aux | $n^s$ | $n^r$ | B-1e-6 | B-1e-5 | C-1e-6 | C-1e-5 | TR1 | Avg-H |
|---|---|---|---|---|---|---|---|---|---|
| DigiFace1M | N/A | 1.2M | 0 | 15.31 | 29.59 | 26.06 | 36.34 | 32.30 | 78.97 |
| RealDigiFace | Y | 1.2M | 0 | 21.37 | 39.14 | 36.18 | 45.55 | 42.64 | 81.34 |
| DCFace | Y | 1.2M | 0 | 22.48 | 47.84 | 35.27 | 58.22 | 45.94 | 91.56 |
| AugGen | N | 0.6M | 0 | **29.40** | **54.54** | **45.15** | **61.52** | 52.33 | 88.78 |
| AugGen Repro | N | 0.6M | 0 | 15.71 | 45.97 | 31.54 | 58.61 | 53.61 | 90.64 |
| CASIA-WebFace | N/A | 0 | 0.5M | 1.02 | 5.06 | 0.73 | 5.37 | 58.12 | 94.21 |
| CASIA-WebFace † | N/A | 0 | 0.5M | 0.74 | 3.94 | 0.38 | 3.92 | **59.64** | **94.84** |
| WebFace160K | N/A | 0 | 0.16M | 32.13 | 72.18 | 70.37 | 78.81 | 61.51 | 92.50 |
| WebFace160K † | N/A | 0 | 0.16M | 34.84 | 74.10 | 72.56 | 81.26 | 62.59 | 93.32 |
| ScoreMix Repro | N | 0.2M | 0 | 28.15 | 57.71 | 54.66 | 67.06 | 56.38 | 92.47 |
| AugGen | N | 0.2M | 0.16M | 34.83 | 76.21 | 75.02 | 82.91 | 61.41 | 93.78 |
| ScoreMix (Ours) | N | 0.2M | 0.16M | **35.95** | **76.41** | **76.45** | **83.58** | **63.09** | **93.87** |

### 4.3 WHICH CLASSES ARE BEST TO MIX?

In this section, we systematically study which classes are best for approaches like AugGen (Rahimi et al., 2025) or our ScoreMix. By "best," we mean that the generated samples using the selected classes deliver the highest performance increase compared to the baseline discriminator. To determine this, we first compare the distances between every pair of classes in (i) the learned condition space of the generator and (ii) the embedding space of the discriminator. More precisely, given $l$ labels in our dataset, we train a discriminator that maps each class to an embedding vector, forming an embedding matrix $E \in \mathbb{R}^{l \times d_E}$ (i.e., the *learned* class centers used for margin losses). Similarly, for each class we have a unique condition vector that is mapped to the hidden latent of the denoiser network, forming a matrix $C \in \mathbb{R}^{l \times d_C}$.

For $E$, since it arises from the discriminator's training, we use *cosine distance* as our metric, which we denote $\text{dist}_{\text{emb}}$. For the condition space $C$, we experiment with two popular metrics, *cosine distance* and *Euclidean (L2) distance*, both denoted $\text{dist}_{\text{cond}}$. This process is depicted in Algorithm 2.

We explore the following hypotheses:

1. Classes that are **closest** in the **embedding** space may be less helpful: because the generator is imperfect, it cannot capture subtle differences between already similar classes, yielding samples that do not challenge the discriminator.

2. Under common metrics (*e.g.*, cosine or L2), interpolating between *closer* conditions may produce better overall samples, potentially improving the discriminator's performance.

3. A combination: select source classes that are both close in the condition space and distant in the embedding space.

For each setting, we select **10K** class pairs and generate **20** samples per pair, matching the size of the original dataset. Results are shown in Table 2. The "Class Sel Mixing Strategy" column indicates how classes were chosen: *Random* (as in (Rahimi et al., 2025)), or based on their distances.

The first key observation is that adding these augmentations increases average discriminator performance by up to 6%, independent of the mixing strategy. To validate hypothesis (1), we compare

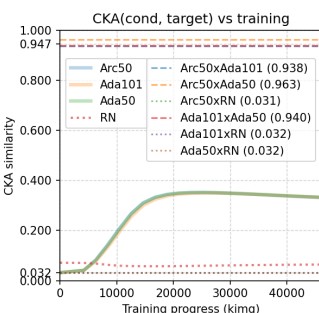 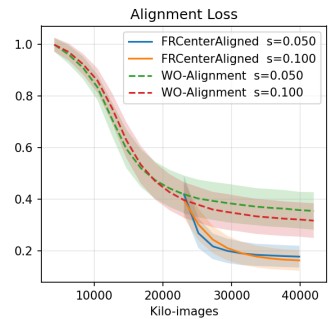 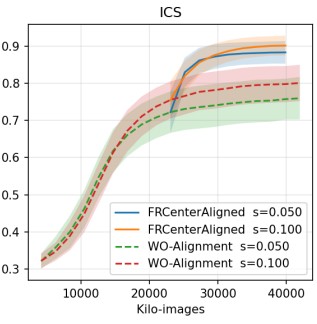

Figure 4: Geometry preservation of various spaces measured using CKA during the training of the generator.

Figure 5: Alignment loss to class-centers before and after applying alignment regularization during the training of the generator.

Figure 6: Intra Class Similarity (ICS) before and after applying alignment regularization during the training of the generator.

strategies based on embedding distances and find that mixing pairs with larger embedding distances yields the greatest gains. In contrast, selecting classes according to condition distances (Close/Dist measured in cosine or $L_2$) has a negligible effect, thereby invalidating hypothesis (2). The critical role of the selection process is also evident from the "Diff." column of Table 2. For instance, when source classes are chosen by their embedding-space distances, the mean pairwise distance is 2.52 (substantially higher than the 0.11 or 0.56 observed in condition space), highlighting the importance of sampling based on the embedding space. (3) Finally advantages of selection based on the two spaces together, as presented in the *Top/Worst Close Cond, Dist Embed*, do not reach the gains achieved through selection based on the embedding space solely.

> **Takeaway:** Choosing the source classes according to their distance in **embedding** space under common distances has more impact on the performance increase of the discriminator. Mixing the **most distant** classes is the most effective class selection strategy for increasing the performance of the discriminator.

**Learned discriminator features as generators condition.** As highlighted previously, under common metrics, there is no clear correspondence between the discriminators' embedding space and the learned generator's condition space, please refer to Appendix Appendix E for details. This gives us the idea to initialize the generator's condition space using the discriminators' class centers and freezing them to observe if we can enforce the missing correspondence. We quickly find that this approach is not feasible, leading to the generators' failure to converge.

> **Takeaway:** Diffusion generators tend not to converge or produce plausible results when we use the highly discriminative features as the frozen conditions.

**Alignment between condition and recognition spaces.** We study whether the generator's *conditional embeddings* preserve the discriminative geometry of a face recognition (FR) backbone. For each training snapshot, we extract one embedding per class from the generator's conditioning module and compare them to the corresponding FR *class centers*. We report two complementary metrics: (i) **Centered Kernel Alignment (CKA)** (Kornblith et al., 2019), which captures global linear relational similarity; and (ii) **CKNNA** (Centered Kernel Nearest-Neighbor Alignment) (Huh et al., 2024), which emphasizes local neighborhood agreement via a soft $k$NN kernel. See Appendix F for their exact definition.

**Interpretation.** Higher values (↑) indicate that the studied spaces are geometrically aligned with, i.e., classwise relations are preserved. Empirically, we observe that phases of training with higher CKA/CKNNA correspond to more stable discriminative performance, supporting a future direction that condition regularization that explicitly encourages recognition-aware geometry.

To test whether alignment is backbone-specific or universal, we are also comparing the condition space against *multiple* recognition models (trained on the same dataset) and treating their class centers as additional anchors. Figure 4 demonstrates consistent alignment across backbones, strengthening the evidence that the generator's conditions capture dataset-intrinsic semantics (note the overlap of

the solid lines). We observe that embeddings from different backbones trained on the same dataset but with distinct loss heads (e.g., Arc/Ada-IR50/100) exhibit highly similar geometric structures (*i.e.*, dashed horizontal lines above 0.9). Their alignments with the condition space are also mutually consistent, although the condition space itself remains significantly farther from the cross alignment of embedding spaces. Since the condition space evolves throughout training, its geometry varies across steps. Nonetheless, it retains some structural similarity to the embedding spaces—unlike a random baseline (RandN), which is a matrix with the same number of rows as Condition Space or Embedding Space and initialized using a normal Gaussian.

Closely related to the nature of how we select the pairs, we introduce the following theorem, which investigates how the pair-wise distances (the selection process of the pairs for mixing) can be preserved in relation to CKA values.

---

**Informal Theorem (CKA and Preservation of Local Geometry)**

Let $\rho = \mathrm{CKA}(X, Y)$ be the centered-kernel alignment between the normalized Grams $\widehat{K}, \widehat{L}$, and let $\Delta_{\widehat{K}} > 0$ denote the (centered, normalized) triplet margin in the reference embedding (Appendix G) and we define $N = \frac{n(n-1)}{2}$. Under the $\widehat{K}$-orthogonal, energy-matched Gaussian misalignment model (Appendix G), the relaxed probability bound that the triplet order is preserved in $Y$ is

$$
\Pr[\Delta_{\widehat{L}} > 0] \geq \Phi\left( \frac{\rho\,\Delta_{\widehat{K}}}{\sqrt{c_{\mathrm{mask}}\,(1-\rho)}} \right), \quad c_{\mathrm{mask}} = \begin{cases} \dfrac{12}{N-1}, & \text{Euclidean squared-distance margins,} \\ \dfrac{2}{N-1}, & \text{cosine-similarity margins.} \end{cases}
$$

which is strictly increasing in $\rho \in (0, 1)$.

---

See Appendix G for a formal statement, proof, and **experimental validation** of the theorem.

**Conjecture 4.2.** As $\mathrm{CKA}(X, Y) \to 1$, the preservation probability approaches 1. Equivalently, in the limit of perfect alignment, almost all local geometric inequalities are preserved.

This highlights that the same observations and methodology can be applied for generating useful samples for the discriminator (*e.g.,* if we have selected the distances based on the other discriminators trained on the same dataset, by changing the loss or backbone), further demonstrating the robustness of the sample selection strategy and its importance.

Table 2: Effect of different strategies for choosing classes to mix for generating augmentations for enhancing the discriminator's performance. Here *Class Sel Mixing Strategy* refers to how we select the classes to mix for the final generation. The Avg column is the average of all reported metrics, for each two rows grouped together (*e.g.*, **Close Embedding Cosine** and **Dist Embedding Cosine** the *Diff* column depicts the absolute difference of the average metrics, presenting the effectiveness of the studied selection strategy.

| Class Sel Mixing Strategy | $n^s$ | $n^r$ | B-1e-6 | B-1e-5 | C-1e-6 | C-1e-5 | TR1 | TR5 | Avg | Diff. |
|---|---|---|---|---|---|---|---|---|---|---|
| WebFace160K | 0 | 0.16M | 33.15 | 72.54 | 70.42 | 78.62 | 61.51 | 66.68 | 63.82 | N/A |
| Random | 0.2M | 0.16M | 34.83 | 76.21 | 75.02 | 82.91 | 61.41 | 66.60 | 66.17 | N/A |
| Close Embedding Cosine | 0.2M | 0.16M | 34.78 | 73.12 | 71.86 | 81.00 | 61.91 | 66.82 | 64.92 | **2.52** |
| Dist Embedding Cosine | 0.2M | 0.16M | 34.42 | **77.46** | **78.62** | **84.04** | **62.66** | 67.46 | **67.44** | |
| Close Condition Cosine | 0.2M | 0.16M | **37.61** | 76.38 | 74.43 | 82.71 | 62.29 | **67.65** | 66.84 | 0.11 |
| Dist Condition Cosine | 0.2M | 0.16M | 34.52 | 77.17 | 76.97 | 83.15 | 62.39 | 67.49 | 66.95 | |
| Close Condition L2 | 0.2M | 0.16M | 37.18 | 72.67 | 72.20 | 80.71 | 62.12 | 66.52 | 65.23 | 0.56 |
| Dist Condition L2 | 0.2M | 0.16M | 33.34 | 75.63 | 75.82 | 82.02 | 61.61 | 66.34 | 65.79 | |
| Top Close Cond, Dist Embed | 0.2M | 0.16M | 34.74 | 76.94 | 76.70 | 83.87 | 62.47 | 67.14 | 66.98 | 1.76 |
| Worst Close Cond, Dist Embed | 0.2M | 0.16M | 33.27 | 74.45 | 74.50 | 81.22 | 61.13 | 66.77 | 65.22 | |
| 3-Plet Sum Max | 0.2M | 0.16M | 31.91 | 74.74 | 74.36 | 81.73 | 63.26 | 68.16 | 65.69 | 1.07 |
| 3-Plet Sum Min | 0.2M | 0.16M | 31.56 | 73.80 | 73.11 | 80.27 | 61.96 | 67.02 | 64.62 | |
| Repro Aligned | 0.2M | 0 | 27.66 | 54.71 | 45.79 | 59.90 | 42.80 | 48.44 | 46.55 | N/A |

## 4.4 BEYOND TWO CLASSES

Here, we study whether we can exploit the gains we observed for more than two classes.

**GPU-accelerated exact extreme $m$-plet mining.** We study the top-$K$ subsets of size $m \in \{3, 4\}$ that optimize a permutation-invariant functional $F$ of the $\binom{m}{2}$ intra-set distances. Naively, $m=3$ requires $\Theta(N^3)$ candidate evaluations (and $m=4$ is $\Theta(N^4)$), which is prohibitive on CPUs even for moderate $N$. Our key observation is that the exhaustive search can be reorganized into *tile-parallel column reductions* that map to high-throughput matrix multiplications and fused argmax/argmin over candidates. This GPU-accelerated approach makes the search feasible even on consumer-level hardware for a moderate $N$ (less than an hour on RTX3090Ti for $m = 3$). To compare across $m$, we report both the sum and its size-invariant version (the mean), i.e., the sum divided by $\binom{m}{2}$ ($= 1$ for pairs, $= 3$ for triples, $= 6$ for quads). In Table 2, we report 3-**plet** Sum/Mean under Min/Max objectives; while $m=3$ improves over the baseline, it does not match the simpler $m=2$ setting. These observations lead us to focus on $m=2$ in the main experiments and not continue with $m=4$ for mixing and training on the 4-plets. See Appendix H for more technical details.

> **Takeaway:** Mixing more than two classes appears to be ineffective in recognition performance with current SOTA diffusion-based generators.

## 4.5 THE MORE ALIGNED, THE BETTER?

As shown in Table 1, training a discriminator on generator reproductions yields lower performance than training on the original dataset. This is expected, since the generator cannot fully capture the fine-grained variations of the real data. To address this, we investigated whether aligning the generator outputs to the discriminator's class centers could help. Figure 5 shows that our regularization indeed improves alignment of generated samples to class centers. However, this comes at the cost of reduced intra-class variability (higher intra-class similarity in Figure 6), which is crucial for capturing identity-preserving information. Consequently, recognition performance on the reproduction set decreases (see the last line of Table 2). Details of the setup, including Coverage and FD metrics and the loss combination with our novel SNR weighting, are provided in Appendix H.

> **Takeaway:** Aligning generator outputs to class centers yields no additional benefit, showing that recognition performance can be achieved without this extra constraint when training on reproduction samples.

## 5 CONCLUSIONS

We have shown that the compositional properties of diffusion model scores can be exploited to substantially improve recognition performance. The approach surpasses the gains from scaling the discriminator capacity and highlights that synthetic augmentation is a more effective alternative. Our analysis further identified which class combinations are most useful for augmentation. Interestingly, we found no clear correlation under standard distance metrics between the generator's condition space and the discriminator's feature space, and forcing the generator to align with class centers during training did not improve discriminator accuracy. To strengthen robustness, we proved that class selection remains stable even under variations in backbone architectures. Finally, we establish a theoretical connection between geometrical alignment metrics (e.g., CKA) and the induced ordering of class pairs, which underpins the stability of our class-mixing strategy to changes in the discriminator selection.

**Limitations.** While our method avoids the need for discriminator-based grid search (unlike AugGen, Rahimi et al. (2025)), it incurs a higher computational sampling cost: generating $m$-plets requires roughly $m$ times the cost of AugGen. This may limit scalability in very large augmentation regimes.

**Future work.** Our findings reveal little correlation between the generator's condition space and the representation space of a strong discriminator. A promising future direction is to investigate whether explicit regularization of the condition space guided by discriminative geometry can improve augmentation quality without sacrificing sample diversity. In particular, exploring representation alignment techniques (e.g., contrastive or CKA-based objectives) may help bridge the gap between generative and discriminative spaces, potentially unlocking further gains in recognition performance.

**Reproducibility statement.** All results in this paper are reproducible, the corresponding code and synthetic datasets will be publicly released.

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

## A    APPENDIX

## B    MORE EXAMPLES ON CHOOSING $\alpha$ AND $\beta$

In the figures Figure 7, Figure 8 and Figure 9 more examples of different values of $\alpha$ and $\beta$ are depicted. For each panel, the ID combinations are fixed across the figures to also highlight the consistency of the IDs with different sources of randomness. Note that the initial value of the seeds was all fixed for each figure to mainly study the effects of mixes of the conditions and the effects of the different values of the $\alpha$ and $\beta$.

## C    ISSUES WITH GENERALIST MODELS

- **License restrictions.** Generalist models like GPT-4o, Gemini (Comanici et al., 2025), or FLUX (Labs, 2024) have restrictive usage policies that prevent their use in sensitive or commercial applications like face recognition.
- **Unknown training data and consent issues.** Many generalist models are trained on private data, where subject consent cannot be guaranteed. This poses a major concern for face recognition systems, medical applications, and other sensitive use cases—an issue our work explicitly avoids.

## D    ALIGNMENT AUGMENTED LOSS

Here, we describe how we applied the alignment loss during training of the generator.

### D.1    PRELIMINARIES: THE EDM2 LOSS FUNCTION

We build upon the uncertainty-aware loss function from the EDM2 (Karras et al., 2024b) framework. At each training step, a noise level $\sigma$ is sampled, and a clean image $\mathbf{X}$ is corrupted to $\mathbf{X}_\sigma$. The network $S(\mathbf{X}, \sigma)$ then predicts the denoised image $\hat{\mathbf{X}}_0$ and a log-variance term $\log(\mathbf{v})$. The loss is evaluated over a distribution of noise levels, training the network to denoise effectively across the entire corruption process:

$$\mathcal{L}_{\text{diff}} = \frac{\sigma^2 + \sigma_{\text{data}}^2}{(\sigma \cdot \sigma_{\text{data}})^2} \cdot \frac{1}{\exp(\log(\mathbf{v}))} \cdot (\hat{\mathbf{X}}_0 - \mathbf{X})^2 + \log(\mathbf{v}) \tag{6}$$

The negative values this loss can produce reflect the model learning to be confident (low predicted $\log(\mathbf{v})$) only when its denoising predictions are accurate.

### D.2    DISCRIMINATOR-GUIDED ALIGNMENT OF THE DENOISING PATH

While $\mathcal{L}_{\text{diff}}$ guides the pixel-level accuracy of the prediction $\hat{\mathbf{X}}_0$ at each step, it does not explicitly enforce its semantic integrity. We introduce an auxiliary loss to align the network's prediction at every timestep with its corresponding class identity.

We denote the $\mathcal{F}_{\text{fr}}(\cdot)$ as the feature extractor from a pre-trained face recognition (FR) model (*i.e.*, usually the $f_{\theta_{\text{dis}}}$ without the classification head). For each class $k$, we pre-compute the class center $\mathbf{c}_k = \mathbb{E}_{\mathbf{X} \sim \text{class}_k}[\mathcal{F}_{\text{fr}}(\mathbf{X})]$. Alternatively, this can also be the class centers of the $f_{\theta_{\text{dis}}}$.d

For a given noisy input $\mathbf{x}_\sigma$ from a sample of class $k$, the diffusion model $D$ predicts the denoised image $\hat{\mathbf{X}}_0 = S(\mathbf{X}_\sigma, \sigma)$. We apply the alignment loss to this prediction:

$$\mathcal{L}_{\text{align}} = 1 - \frac{\mathcal{F}_{\text{fr}}(S(\mathbf{X}_\sigma, \sigma)) \cdot \mathbf{c}_k}{\|\mathcal{F}_{\text{fr}}(S(\mathbf{X}_\sigma, \sigma))\| \|\mathbf{c}_k\|} \tag{7}$$

This loss acts as a semantic gradient, pulling the network's prediction at each step towards the correct identity manifold, thereby guiding the entire denoising path.

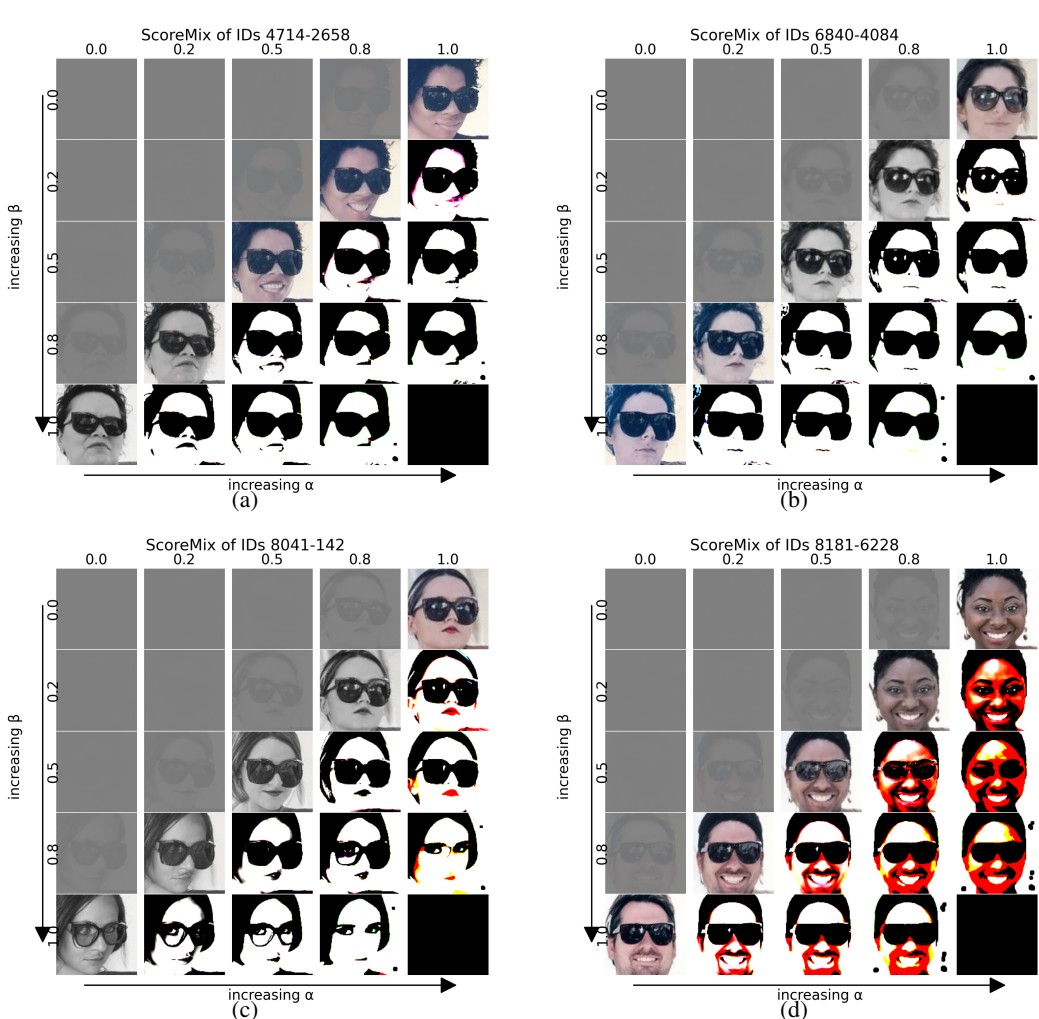

Figure 7: Effect of mixing scores in SCOREMIX. Sub-figures c, d show the images obtained for four different input pairs while sweeping the mixing coefficients $\alpha$ (horizontal axis, *increasing left → right*) and $\beta$ (vertical axis, *increasing top → bottom*). All randomness aspects were fixed. All images were generated by fixing all the seeds to the initial value of '0'.

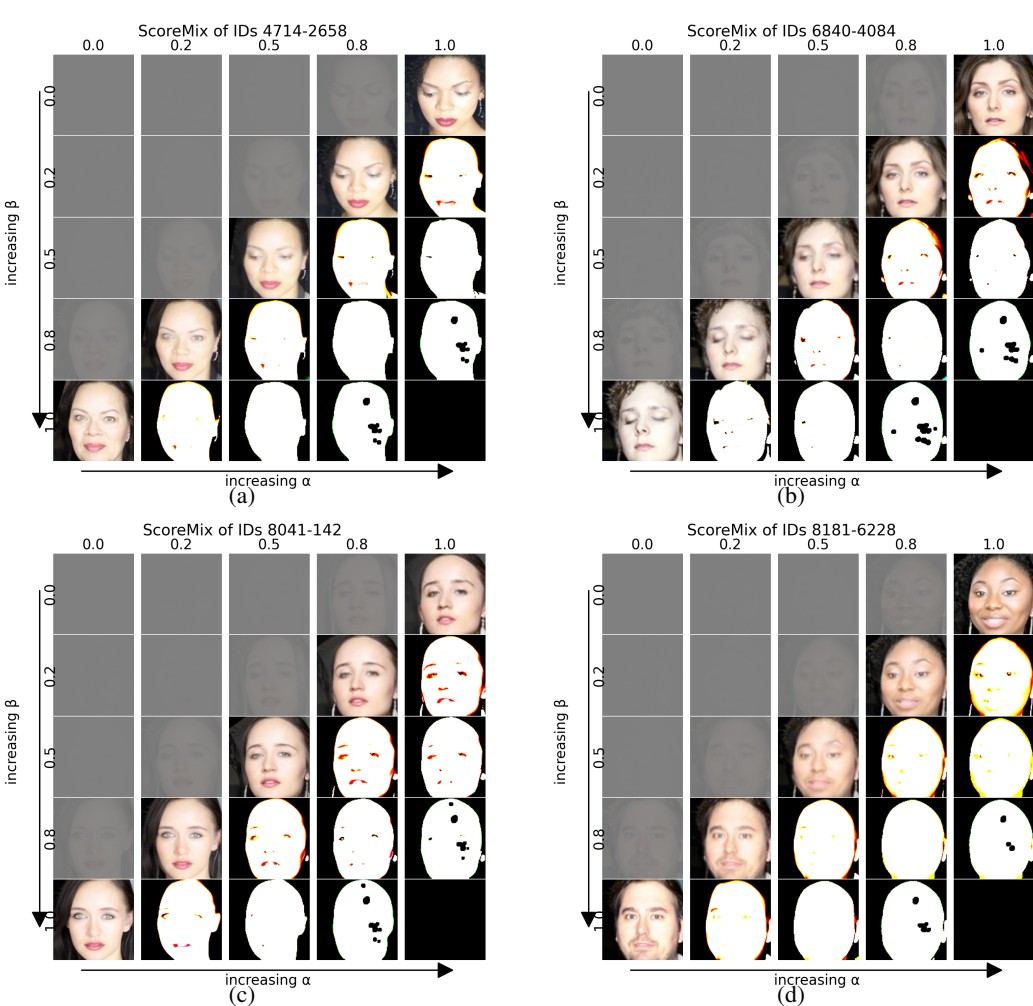

Figure 8: Effect of mixing scores in SCOREMIX. Sub-figures a–d show the images obtained for four different input pairs while sweeping the mixing coefficients $\alpha$ (horizontal axis, *increasing left → right*) and $\beta$ (vertical axis, *increasing top → bottom*). All randomness aspects were fixed. All images were generated by fixing all the seeds to the initial value of '1'.

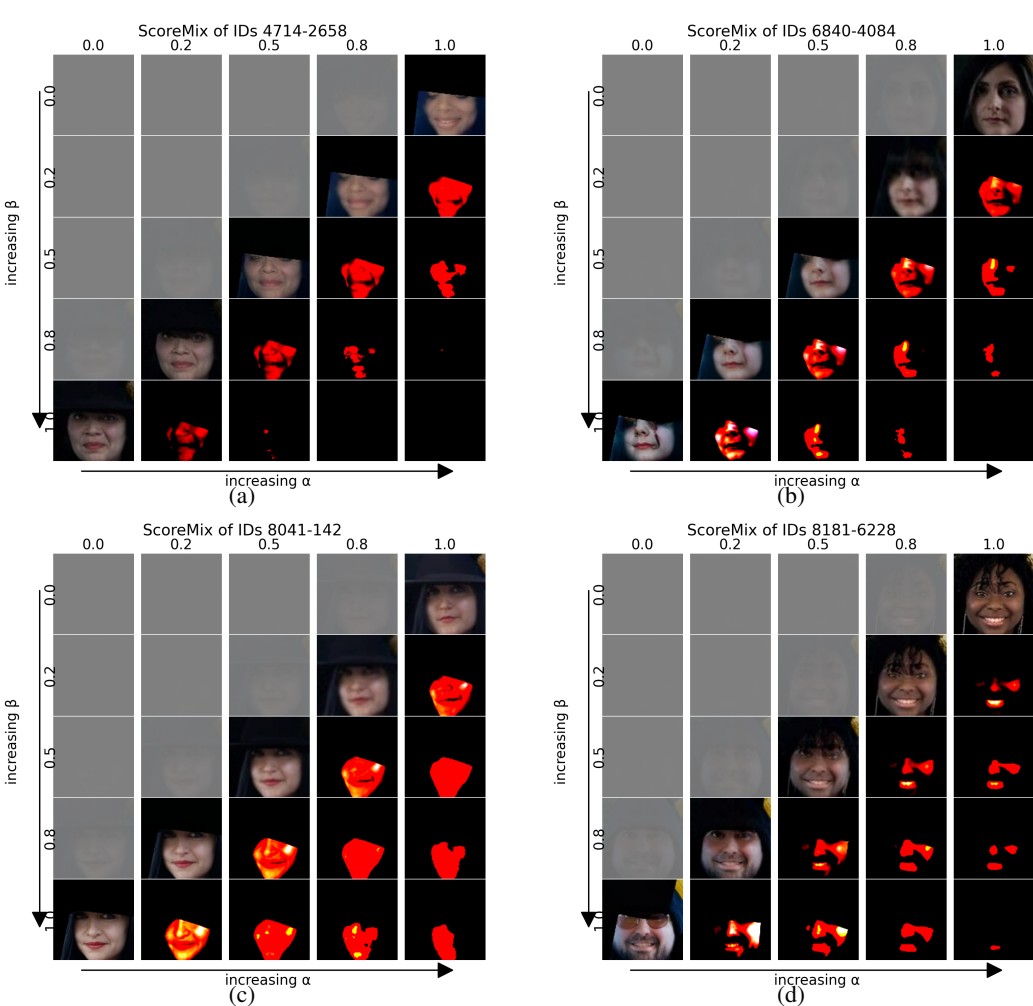

Figure 9: Effect of mixing scores in SCOREMIX. Sub-figures a–d show the images obtained for four different input pairs while sweeping the mixing coefficients $\alpha$ (horizontal axis, *increasing left → right*) and $\beta$ (vertical axis, *increasing top → bottom*). All randomness aspects were fixed. All images were generated by fixing all the seeds to the initial value of '6'.

### D.3 NOISE-AWARE LOSS WEIGHTING

The core challenge in applying this auxiliary loss is that the prediction, $\hat{\mathbf{X}}_0 = \mathrm{S}(\mathbf{X}_\sigma, \sigma)$, is an *estimate* whose reliability is a direct function of the noise level $\sigma$. At high noise levels (low Signal-to-Noise Ratio), this prediction is a high-variance estimate. Enforcing a strict feature-space constraint on such a high-variance prediction can introduce conflicting gradients and destabilize training. Conversely, at low noise levels (high SNR), the prediction is a much more reliable, lower-variance estimate, making it an ideal target for semantic guidance.

We therefore modulate the alignment loss with a dynamic, SNR-aware weight $w_{\mathrm{snr}}(\sigma)$ that scales the loss based on the reliability of the prediction:

$$w_{\mathrm{snr}}(\sigma) = \exp(-k \cdot \sigma^2) \tag{8}$$

where $k$ is a hyperparameter. This weighting scheme ensures that the semantic guidance from $\mathcal{L}_{\mathrm{align}}$ is applied most strongly only when the model's denoised prediction is coherent and meaningful.

### D.4 CURRICULUM FOR STABLE ALIGNMENT

To further stabilize training, especially in the initial phases where the generator is still learning basic image structures, we introduce the alignment loss gradually. We define a start point, $n_{\mathrm{start}}$, and a ramp-up duration, $n_{\mathrm{ramp}}$, measured in training images. The curriculum weight, $w_{\mathrm{ramp}}$, scales the influence of the alignment loss based on the current training progress, $n_{\mathrm{cur}}$:

$$w_{\mathrm{ramp}} = \min\left(\max\left(0, \frac{n_{\mathrm{cur}} - n_{\mathrm{start}}}{n_{\mathrm{ramp}}}\right), 1.0\right) \tag{9}$$

This allows the network to first learn basic image synthesis before being gently steered by the alignment objective.

### D.5 FINAL LOSS FORMULATION

Our final training objective is the expectation over the data distribution and noise levels, combining all components to guide the entire denoising trajectory towards producing semantically and visually accurate results:

$$\mathcal{L}_{\mathrm{total}} = \mathbb{E}_{\mathbf{X},k,\sigma}\left[\mathcal{L}_{\mathrm{diff}} + \lambda \cdot w_{\mathrm{ramp}} \cdot w_{\mathrm{snr}}(\sigma) \cdot \mathcal{L}_{\mathrm{align}}\right] \tag{10}$$

where $\lambda$ is a scalar hyperparameter balancing the two objectives. This formulation provides a stable and principled method for training a diffusion generator that is guided by semantic constraints at every step of the generation process.

### D.6 EVALUATION METRICS

We evaluate identity fidelity and intra-class diversity in the feature space of a frozen face-recognition (FR) model, $\mathcal{F}_{\mathrm{fr}}(\cdot)$, trained on the target domain. Let $G(z, k)$ be the image for class $k$ and seed $z$, and $S_k = \{z_1, \ldots, z_N\}$ a fixed set of $N$ seeds per class (kept constant across runs).

**Feature normalization.** All feature vectors and class centers are $\ell_2$-normalized prior to computing any metric. Denote $\mathbf{f}_{i,k} = \mathrm{norm}(\mathcal{F}_{\mathrm{fr}}(G(z_i, k)))$ and $\mathbf{c}_k^{\mathrm{target}} = \mathrm{norm}(\text{center from real data})$. We compute the (pre-)centroid $\tilde{\mathbf{c}}_k^{\mathrm{gen}} = \frac{1}{N}\sum_{i=1}^N \mathbf{f}_{i,k}$ and then re-normalize $\mathbf{c}_k^{\mathrm{gen}} = \mathrm{norm}(\tilde{\mathbf{c}}_k^{\mathrm{gen}})$. With unit-norm vectors, the cosine distance reduces to $d_{\mathrm{cos}}(\mathbf{a}, \mathbf{b}) = 1 - \mathbf{a}^\top \mathbf{b}$.

**Alignment Loss to Target Center (Fidelity).** Average cosine distance of samples to the real class center, this is the same as it being reported in the paper (lower is better):

$$\mathcal{M}_{\mathrm{align}}(k) = \frac{1}{N}\sum_{i=1}^N d_{\mathrm{cos}}(\mathbf{f}_{i,k}, \mathbf{c}_k^{\mathrm{target}}).$$

**Intra-Class Cosine Similarity (Diversity).** Average cosine similarity of samples of the same class to the generated centroid (lower is better):

$$\mathcal{M}_{\mathrm{ICS}}(k) = \frac{1}{N}\sum_{i=1}^N 1 - d_{\mathrm{cos}}(\mathbf{f}_{i,k}, \mathbf{c}_k^{\mathrm{gen}}).$$

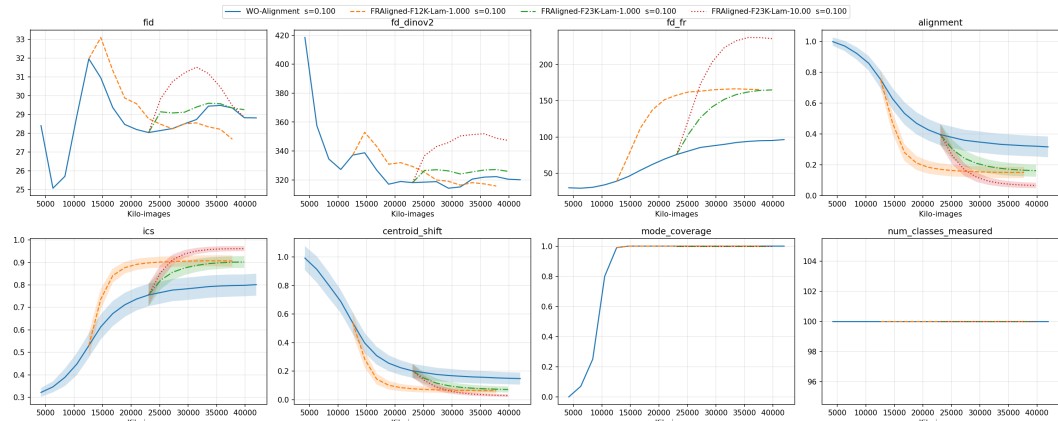

Figure 10: Effect of alignment regularization on various metrics during the training of the diffusion-based generator.

**Centroid Shift (Bias).** Cosine distance between generated and target centers (lower is better):

$$\mathcal{M}_{\text{shift}}(k) = d_{\cos}(\mathbf{c}_k^{\text{gen}}, \mathbf{c}_k^{\text{target}}).$$

**Mode Coverage.** Fraction of evaluated classes whose generated centroid is nearest (by cosine similarity) to their own target center among the evaluated subset (higher is better):

$$\mathcal{M}_{\text{coverage}} = \frac{1}{|K_{\text{eval}}|} \sum_{k \in K_{\text{eval}}} \mathbb{I}\left[k = \arg\max_{j \in K_{\text{eval}}} \mathbf{c}_k^{\text{gen} \top} \mathbf{c}_j^{\text{target}}\right].$$

(If target centers for all classes are available and you want a stricter criterion, replace $K_{\text{eval}}$ with $K_{\text{all}}$ above.) We report mean $\pm$ standard deviation across $k \in K_{\text{eval}}$ for distance-based metrics.

**FD.** We also report Frechet Distance (FD), under various backbones, like InceptionV3 (Szegedy et al., 2016), DINOv2 (Oquab et al., 2023), and also using the embeddings of the same discriminator denoted as $\text{FD}_{\text{FR}}$.

We show the results in Figure 10. Here, we observe that light regularization for alignment tends to converge to similar values whether it is applied early or later (*i.e.*, note where the orange and green dashed lines end for both the **ICS** and **Alignment Loss**, with the orange plot demonstrating much earlier regularization). We also observe that although the Alignment Loss is decreasing, the **ICS** is increasing, which causes the generated images to appear less diverse. We believe this is the main reason why reproduction with a more aligned generator penalizes the downstream performance of the discriminator on the reproduction dataset. Additionally, as highlighted in earlier works (Stein et al., 2023), FD does not correlate well with sample quality and downstream performance (Rahimi et al., 2025). In contrast, $\text{FD}_{\text{DINOv2}}$ better captures this correlation. Moreover, highly discriminative features (*e.g.,* FR features) also do not appear well suited for reporting sample quality.

## E  ILLUSTRATION OF EMBEDDING AND CONDITION SPACE

After normalizing and identifying the most similar pairs (i.e., those with cosine distance 0, or equivalently, cosine similarity 1), we shift these zero distances to –1 to improve visual contrast. The resulting distance matrices for all sample pairs, $\mathbf{E}$ and $\mathbf{C}$, are shown in Figure 11. From these plots, we see no obvious correlation between the two spaces.

As another way of viewing this, if we flatten the matrices and use a few pairs like a set $\mathcal{S}$:

$$\mathcal{S} \subseteq \{1, \ldots, 10000\} \times \{1, \ldots, 10000\}, \qquad (i, j) \in \mathcal{S}.$$

And treating the distances as a 1D signal where each tick of the x-axis corresponds to a unique combination of $i$ and $j$, we get a plot like Figure 12. Here, the red vertical lines are illustrating when

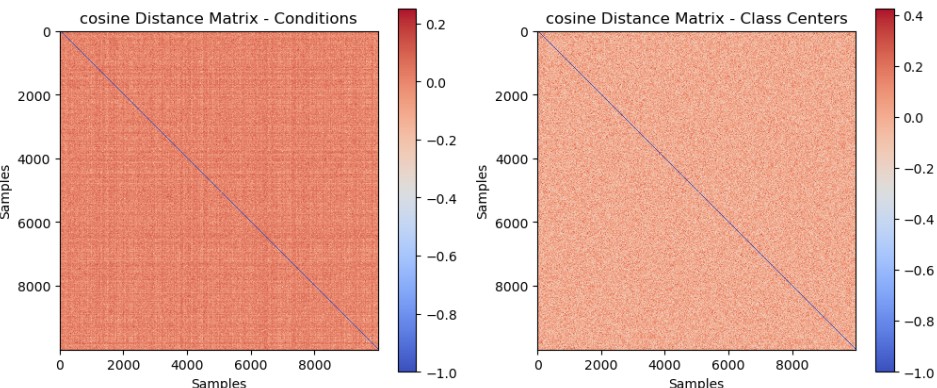

Figure 11: Shifted Matrix Cosine Matrix Distances between each pair in the condition and embedding space.

the both condition and embedding space are having a distance lower than $0.4$, We also apply some peak detection especially for the embedding space as we demonstrated the more distant we have in the embedding space the more beneficial the synthetic samples will be. Here, we again observe that these two spaces do not correlate well.

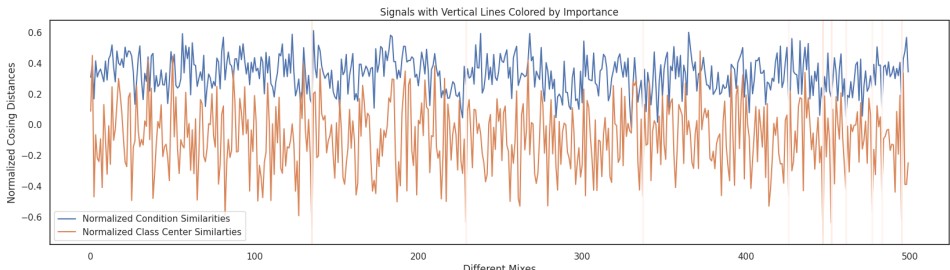

Figure 12: Selected few distances in condition and embedding space. Here x-axis depicts a few unique combinations of classes.

## F    ALIGNMENT METRICS AND EVALUATION PROTOCOL

**Linear CKA (Kornblith et al., 2019).** Given representations $X \in \mathbb{R}^{n \times d_x}$ and $Y \in \mathbb{R}^{n \times d_y}$ for the same $n$ items (like center classes), let

$$H = I_n - \tfrac{1}{n}\mathbf{1}\mathbf{1}^\top, \quad K_X = HXX^\top H, \quad K_Y = HYY^\top H.$$

Where the $\mathbf{1}$ is all one vector of size $n$ and $I_n$ is the identity matrix. The (linear) CKA similarity is

$$\mathrm{CKA}(X,Y) \;=\; \frac{\langle K_X, K_Y \rangle_F}{\|K_X\|_F \|K_Y\|_F} \;=\; \frac{\|X^\top Y\|_F^2}{\|X^\top X\|_F \|Y^\top Y\|_F}.$$

Values near 1 indicate strong global relational alignment; values near 0 indicate weak or no alignment.

**CKNNA (Huh et al., 2024).** CKNNA measures *local* (neighborhood) alignment. For a temperature $\tau > 0$, define a soft neighbor kernel on $X$:

$$A_X(i,j) \;=\; \begin{cases} \dfrac{\exp\left(\langle \hat{x}_i, \hat{x}_j \rangle / \tau\right)}{\sum_{k \neq i} \exp\left(\langle \hat{x}_i, \hat{x}_k \rangle / \tau\right)} & \text{if } i \neq j, \\ 0 & \text{if } i = j, \end{cases} \quad \hat{x}_i = \frac{x_i}{\|x_i\|_2},$$

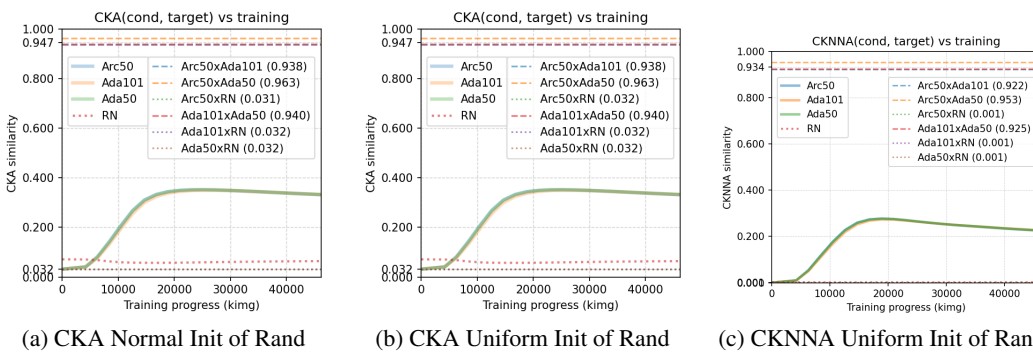

(a) CKA Normal Init of Rand    (b) CKA Uniform Init of Rand    (c) CKNNA Uniform Init of Rand

Figure 13: CKA/CKNNA plots under different random initialization schemes.

and analogously $A_Y$ for $Y$. Using centered versions $\tilde{A}_X = HA_XH$ and $\tilde{A}_Y = HA_YH$, the (cosine-type) CKNNA similarity is

$$\text{CKNNA}(X,Y) = \frac{\langle \tilde{A}_X, \tilde{A}_Y \rangle_F}{\|\tilde{A}_X\|_F \|\tilde{A}_Y\|_F}.$$

Smaller $\tau$ emphasizes sharper, more discrete neighborhoods; larger $\tau$ yields smoother neighborhoods. Higher values indicate better agreement of local neighborhoods across spaces. We set $X = generator$ $condition\ embeddings$ (one per class) and $Y = FR\ class\ centers$. We report $\text{CKA}(X,Y)$ and $\text{CKNNA}(X,Y)$ as similarities in $[0,1]$ (higher is better). Intuitively, CKA captures global relational structure, while CKNNA emphasizes whether each class's nearest neighbors (by angular similarity) are consistent across the two spaces. Repeat the above with centers from several recognition models trained on the same dataset (e.g., ArcFace, AdaFace). Consistently high alignment across backbones implies that the learned embedding space captures highly similar data representation spaces.

## G    FULL VERSION AND PROOF OF THEOREM 4.3

**Theorem G.1** (CKA and local-order preservation under $\widehat{K}$-orthogonal, energy-matched Gaussian misalignment). *Let $X, Y \in \mathbb{R}^{n \times d}$ and define the centered Gram matrices*

$$K = HXX^\top H, \qquad L = HYY^\top H, \tag{11}$$

*with $H = I - \frac{1}{n}\mathbf{1}\mathbf{1}^\top$. Normalize $\widehat{K} := K/\|K\|_F$, $\widehat{L} := L/\|L\|_F$, and define the (linear) CKA*

$$\rho := \left\langle \widehat{K}, \widehat{L} \right\rangle_F \in [0,1). \tag{12}$$

*For distinct indices $(i, j, k)$, define the squared-Euclidean triplet mask $T_{i;jk} \in \mathbb{S}^n$ by*

$$(T_{i;jk})_{jj} = +1, \quad (T_{i;jk})_{kk} = -1, \quad (T_{i;jk})_{ij} = (T_{i;jk})_{ji} = -1, \quad (T_{i;jk})_{ik} = (T_{i;jk})_{ki} = +1, \tag{13}$$

*and 0 elsewhere. Let $\mathcal{S}_c := \{M \in \mathbb{S}^n : M\mathbf{1} = \mathbf{0}\}$ and $N := \dim(\mathcal{S}_c) = \frac{n(n-1)}{2}$ [1]. Let $T_c := HT_{i;jk}H \in \mathcal{S}_c$ (so $\|T_c\|_F \leq \|T_{i;jk}\|_F = \sqrt{6}$). Define the centered, normalized triplet margins*

$$\Delta_{\widehat{K}} := \left\langle T_c, \widehat{K} \right\rangle_F, \qquad \Delta_{\widehat{L}} := \left\langle T_c, \widehat{L} \right\rangle_F. \tag{14}$$

*Assume the following misalignment model on the Hilbert space $(\mathcal{S}_c, \langle \cdot, \cdot \rangle_F)$:*

*1. $\widehat{L} = \rho\widehat{K} + E$ with $\left\langle E, \widehat{K} \right\rangle_F = 0$ (orthogonal decomposition);*

*2. $E$ is a zero-mean Gaussian random element supported on $\{\widehat{K}\}^\perp \cap \mathcal{S}_c$ that is isotropic on that $(N-1)$-dimensional slice: its covariance is $\sigma^2 I$;*

---

[1] Note that $\dim\{M\} = \dim\{M^\top\} = n(n+1)/2$. The centering map $M \to M\mathbf{1}$ has rank $n$ on $\mathbb{S}^n$, so $\dim(\mathcal{S}_c) = n(n+1)/2 - n = n(n-1)/2$.

3. *the variance level is energy matched,*

$$\sigma^2 = \frac{1-\rho^2}{N-1}, \tag{15}$$

*which yields $\mathbb{E}\|E\|_F^2 = 1 - \rho^2$.*

*Then, for any triplet with $\Delta_{\widehat{K}} > 0$,*

$$\mathbb{P}\big[\Delta_{\widehat{L}} > 0\big] = \Phi\left(\frac{\rho\,\Delta_{\widehat{K}}\,\sqrt{N-1}}{\|\Pi_\perp T_c\|_F\,\sqrt{1-\rho^2}}\right), \qquad \Pi_\perp T_c := T_c - \big\langle T_c, \widehat{K}\big\rangle_F\,\widehat{K}, \tag{16}$$

*where $\Phi$ is the standard normal CDF. The right-hand side is strictly increasing in $\rho \in [0, 1)$, and by continuity the $\rho \to 1$ limit equals 1.*[2]

*Proof.* All inner products and norms are Frobenius on $\mathcal{S}_c$. By the model, $\widehat{L} = \rho\,\widehat{K} + E$ with $E \in \{\widehat{K}\}^\perp$ a.s. For the fixed triplet, define the continuous linear functional $\Delta(\cdot) := \langle T_c, \cdot\rangle_F$. Then

$$\Delta_{\widehat{L}} = \Delta(\widehat{L}) = \rho\,\Delta(\widehat{K}) + \Delta(E) = \rho\,\Delta_{\widehat{K}} + \langle T_c, E\rangle_F = \rho\,\Delta_{\widehat{K}} + \langle \Pi_\perp T_c, E\rangle_F, \tag{17}$$

since $E \in \{\widehat{K}\}^\perp$. By Gaussianity and isotropy on the slice,

$$\langle \Pi_\perp T_c, E\rangle_F \sim \mathcal{N}\Big(0,\ \sigma^2\,\|\Pi_\perp T_c\|_F^2\Big), \quad \sigma^2 = \frac{1-\rho^2}{N-1}. \tag{18}$$

Hence $\Delta_{\widehat{L}} \sim \mathcal{N}\Big(\rho\,\Delta_{\widehat{K}},\ \frac{1-\rho^2}{N-1}\,\|\Pi_\perp T_c\|_F^2\Big)$, and threfore

$$\mathbb{P}\big[\Delta_{\widehat{L}} > 0\big] = \Phi\left(\frac{\rho\,\Delta_{\widehat{K}}}{\sigma\,\|\Pi_\perp T_c\|_F}\right) = \Phi\left(\frac{\rho\,\Delta_{\widehat{K}}\,\sqrt{N-1}}{\|\Pi_\perp T_c\|_F\,\sqrt{1-\rho^2}}\right), \tag{19}$$

which is equation 16. Monotonicity follows since $f(\rho) := \rho/\sqrt{1-\rho^2}$ has $f'(\rho) = (1-\rho^2)^{-3/2} > 0$ on $(0, 1)$ and is continuous at 0, and $\Phi$ is increasing. $\qquad\square$

**Corollary G.2** (Unnormalized form). *With $\Delta_K := \langle T_c, K\rangle_F = \|K\|_F\,\Delta_{\widehat{K}}$ and $\Delta_L := \langle T_c, L\rangle_F$, we have*

$$\mathbb{P}\big[\Delta_L > 0\big] = \Phi\left(\frac{\rho\,\Delta_K\,\sqrt{N-1}}{\|K\|_F\,\|\Pi_\perp T_c\|_F\,\sqrt{1-\rho^2}}\right). \tag{20}$$

**Corollary G.3** (Universal lower bound). *For $\rho \in [0, 1]$, using $\|\Pi_\perp T_c\|_F \leq \|T_c\|_F \leq \sqrt{6}$ and $1 - \rho^2 \leq 2(1 - \rho)$,*

$$\mathbb{P}\big[\Delta_{\widehat{L}} > 0\big] \geq \Phi\left(\frac{\rho\,\Delta_{\widehat{K}}}{\sqrt{\frac{12}{N-1}\,(1-\rho)}}\right). \tag{21}$$

**Remark 1** (On centering and the choice of $T_c$). *For any $K \in \mathcal{S}_c$ and any $T \in \mathbb{R}^{n \times n}$, $\langle T, K\rangle_F = \langle HTH, K\rangle_F$ because $HK = KH = K$. Thus replacing $T_{i;jk}$ by $T_c = HT_{i;jk}H$ does not change triplet margins against centered Grams, and ensures $T_c \in \mathcal{S}_c$. Moreover, $\|T_{i;jk}\|_F^2 = 6$ and $H$ is a contraction in Frobenius norm, so $\|T_c\|_F \leq \sqrt{6}$.*

**Remark 2** (Alternative residual model). *If $E$ is uniformly distributed on the Frobenius sphere of radius $\sqrt{1-\rho^2}$ in $\{\widehat{K}\}^\perp$, then $\langle \Pi_\perp T_c, E\rangle_F$ has the 1D marginal of a random point on that sphere (symmetric Beta-type law). The exact Normal tail in equation 16 is then replaced by the corresponding spherical CDF. Note that corollary G.3 remain valid lower bounds.*

---

[2]When $\rho = 1$, we have $\sigma^2 = 0$ so $E \equiv 0$ and hence $\Delta_{\widehat{L}} = \Delta_{\widehat{K}} > 0$ deterministically. Therefore, $\mathbb{P}[\Delta_{\widehat{L}} > 0] = 1$, which also matches the limit of equation 16 as $\rho \uparrow 1$.

## G.1 COSINE DISTANCE AND KERNEL-INDUCED DISSIMILARITIES

**Corollary G.4** (Cosine similarity case: exact bound and universal lower bound). *Let $\widetilde{X}$ and $\widetilde{Y}$ be the row-normalized versions of $X$ and $Y$ (each row scaled to unit $\ell_2$ norm). Define the centered cosine-similarity Gram matrices $S := H\,\widetilde{X}\widetilde{X}^\top H$, $R := H\,\widetilde{Y}\widetilde{Y}^\top H$, their normalizations $\widehat{S} := S/\|S\|_F$, $\widehat{R} := R/\|R\|_F$, and $\rho_{\cos} := \langle \widehat{S}, \widehat{R} \rangle_F \in [-1, 1]$. For a triplet $(i, j, k)$, define the cosine-margin functional $\Delta^{\cos}(M) := M_{ij} - M_{ik}$ via the symmetric mask*

$$(T^{\cos}_{i;jk})_{ij} = (T^{\cos}_{i;jk})_{ji} = +\tfrac{1}{2}, \qquad (T^{\cos}_{i;jk})_{ik} = (T^{\cos}_{i;jk})_{ki} = -\tfrac{1}{2}, \qquad else\ 0,$$

*so that $\Delta^{\cos}(M) = \langle T^{\cos}_{i;jk}, M \rangle_F$ for any symmetric $M$ and $\|T^{\cos}_{i;jk}\|^2_F = 1$. Let $T^{\cos}_c := H\,T^{\cos}_{i;jk}\,H \in \mathcal{S}_c$, and set*

$$\Delta^{\cos}_{\widehat{S}} := \langle T^{\cos}_c, \widehat{S} \rangle_F, \qquad \Delta^{\cos}_{\widehat{R}} := \langle T^{\cos}_c, \widehat{R} \rangle_F, \qquad \Pi_\perp T^{\cos}_c := T^{\cos}_c - \langle T^{\cos}_c, \widehat{S} \rangle_F\, \widehat{S}. \tag{22}$$

*Under the $\widehat{S}$-orthogonal, energy-matched Gaussian isotropy model from Theorem G.1 with $N = \dim(\mathcal{S}_c) = \frac{n(n-1)}{2}$ and $\sigma^2 = (1 - \rho^2_{\cos})/(N - 1)$, for any triplet with $\Delta^{\cos}_{\widehat{S}} > 0$ we have the exact identity*

$$\mathbb{P}\big[\Delta^{\cos}_{\widehat{R}} > 0\big] = \Phi\left(\frac{\rho_{\cos}\,\Delta^{\cos}_{\widehat{S}}\,\sqrt{N-1}}{\|\Pi_\perp T^{\cos}_c\|_F\,\sqrt{1 - \rho^2_{\cos}}}\right). \tag{23}$$

*Moreover, since $\|\Pi_\perp T^{\cos}_c\|_F \le \|T^{\cos}_c\|_F \le 1$ and $1 - \rho^2_{\cos} \le 2(1 - \rho_{\cos})$ for $\rho_{\cos} \in [0, 1]$, we obtain the universal lower bound*

$$\mathbb{P}\big[\Delta^{\cos}_{\widehat{R}} > 0\big] \ge \Phi\left(\frac{\rho_{\cos}\,\Delta^{\cos}_{\widehat{S}}}{\sqrt{\frac{2}{N-1}\left(1 - \rho_{\cos}\right)}}\right). \tag{24}$$

*Equivalently, for cosine distance $d_{\cos}(i, j) = 1 - \cos(i, j)$ the event $d_{\cos}(i, j) < d_{\cos}(i, k)$ is the same as $\Delta^{\cos}(S) > 0$, so equation 23-equation 24 apply unchanged.*

*Proof.* Apply Theorem G.1 with $K \leftarrow S$, $L \leftarrow R$ and $T_c \leftarrow T^{\cos}_c$. The mask norm satisfies $\|T^{\cos}_{i;jk}\|^2_F = 4 \cdot (1/2)^2 = 1$, hence $\|T^{\cos}_c\|_F \le 1$. Since $E \in \{\widehat{S}\}^\perp$ a.s., the variance of $\langle T^{\cos}_c, E \rangle_F$ equals $\sigma^2\|\Pi_\perp T^{\cos}_c\|^2_F$, which gives equation 23; the lower bound follows by the two inequalities above and the monotonicity of $\Phi$. □

**Corollary G.5** (Kernel-induced triplet margins). *Let $k$ be PSD with centered Grams $G^X := HK(X)H$, $G^Y := HK(Y)H$ and $\widehat{G}^X, \widehat{G}^Y$ their normalizations. If a triplet margin admits the linear form $\Delta^k(M) = \langle T^k_{i;jk}, M \rangle_F$ with $T^k_{i;jk} \in \mathcal{S}_c$, then with $T^k_c := HT^k_{i;jk}H$, $\rho_k := \langle \widehat{G}^X, \widehat{G}^Y \rangle_F$, and $\Pi_\perp T^k_c := T^k_c - \langle T^k_c, \widehat{G}^X \rangle_F\, \widehat{G}^X$,*

$$\mathbb{P}\big[\langle T^k_c, \widehat{G}^Y \rangle_F > 0\big] = \Phi\left(\frac{\rho_k\,\langle T^k_c, \widehat{G}^X \rangle_F\,\sqrt{N-1}}{\|\Pi_\perp T^k_c\|_F\,\sqrt{1 - \rho^2_k}}\right), \tag{25}$$

$$\mathbb{P}\big[\langle T^k_c, \widehat{G}^Y \rangle_F > 0\big] \ge \Phi\left(\frac{\rho_k\,\langle T^k_c, \widehat{G}^X \rangle_F}{\sqrt{\frac{\|T^k_c\|^2_F}{N-1}\left(1 - \rho_k\right)}}\right), \tag{26}$$

*under the same isotropic Gaussian misalignment model (and its universal relaxation), respectively.*

## G.2 EXPERIMENTAL VALIDATION

We now verify that the simplified universal lower probability bound is consistent with empirical order preservation across different embedding spaces. For each pair of spaces, we measured the top-$K$ set overlap between different spaces like Arc/Ada-IR50/IR101 (note that we wanted to see if, with these observations, we can verify that higher alignment preserves the ordering and hence the mix selection procedure), Jaccard similarity, and average rank gaps. We also computed the bound-based probability using $\Delta_K \approx 10^{-5}$ as the effective margin (note that $\text{Gap}_A$ and $\text{Gap}_B$ columns) was about this range. If we define the lower bound of Equation 23 as $p_{\text{lower}-\text{bound}}$, multiplying $p_{\text{lower}-\text{bound}}$

Table 3: Empirical vs. bound-based overlap at $K = 20{,}000$. Overlap and Jaccard are computed directly from top-$K$ sets. $p_{\text{lower-bound}}$ is the probability from the practical bound with measured CKA values. "Expected" is $K \cdot p_{\text{lower-bound}}$.

| Pair ($A$–$B$) | CKA | Overlap | Jaccard | $p_{\text{lower-bound}}$ | Exp. | $\text{Gap}_A, \text{Gap}_B$ |
|---|---|---|---|---|---|---|
| ArcIR50–AdaIR50 | 0.9633 | 11885 | 0.423 | 0.599 | 11984 | $7.7 \times 10^{-6}, 9.5 \times 10^{-6}$ |
| AdaIR50–AdaIR101 | 0.9396 | 9869 | 0.328 | 0.575 | 11515 | $9.5 \times 10^{-6}, 1.0 \times 10^{-5}$ |
| ArcIR50–AdaIR101 | 0.9375 | 9658 | 0.318 | 0.574 | 11487 | $7.7 \times 10^{-6}, 1.0 \times 10^{-5}$ |

by $K = 20{,}000$ gives a predicted rough estimation of the overlap that closely matches the observed values.

The bound consistently predicts overlaps of the right order of magnitude, with deviations of $\approx$ 5–10% that are expected due to finite-sample effects and the coarse margin choice. Importantly, the relative ranking across pairs (higher CKA $\Rightarrow$ higher overlap/Jaccard) is preserved, supporting the validity of the bound as a practical predictor of order preservation.

## H ALGORITHMIC DESIGN FOR EXACT EXTREME $m$-PLETS

**Problem and scoring.** Given embeddings $X \in \mathbb{R}^{N \times D}$ and a distance $d(\cdot, \cdot)$, we seek top-$K$ sets $S$ of size $m$ maximizing or minimizing a symmetric functional $F$ of the $\binom{m}{2}$ pairwise distances within $S$. Examples include SUM, MEAN, STD, and order statistics of the pairwise distances; our reducers treat $F$ generically.

**Pairs ($m{=}2$), exact.** We partition the strict upper triangle into $B \times B$ blocks, evaluate a distance block, mask $i \geq j$, and maintain on-device top-$K$ for nearest and farthest pairs. Block size $B$ is chosen experimentally with monitoring the GPU power usage by a simple memory budget to maximize arithmetic intensity while keeping working buffers subquadratic.

**Triples ($m{=}3$), column-exact with global top-$K$.** We tile indices $I$ and $J$ with sizes $(T_i, T_j)$ and traverse their Cartesian product. Within each $(I, J)$ tile, a sub-batch of $P_c$ pair-columns $(i, j)$ is processed as follows:

1. Compute the base pair distances $d_{ij}$ for the $P_c$ columns.
2. Form two candidate matrices $A = X X_I^\top \in \mathbb{R}^{N \times P_c}$ and $B = X X_J^\top \in \mathbb{R}^{N \times P_c}$.
3. For each column $c$ (a fixed $(i, j)$), evaluate $F(\{d_{ij}[c], d_{ik}, d_{jk}\})$ *for all $k \in [N] \setminus \{i, j\}$* via a fused reduction over the $k$-dimension, and select the exact argmax/argmin $k^\star$.
4. Push the resulting triple $(i, j, k^\star)$ and its score to a global device top-$K$.

This procedure is *exact per column*. Global top-$K$ is exact provided at most one $k$ per $(i, j)$ lies above the $K$-th frontier; if necessary, emitting the top-$M$ candidates per column and performing a $K$-way merge yields full exactness (in practice, $M{=}1$ sufficed under our settings). Arithmetic remains $\Theta(N^3)$ but is streamed through GEMM-like blocks; peak memory is $O(N P_c)$, independent of the total number of columns processed.

**Quads ($m{=}4$), per-triple exact greedy expansion.** Given a triple $(i, j, k)$, we evaluate all candidates $l \in [N] \setminus \{i, j, k\}$ in one batched pass by forming the six pairwise distances within $\{i, j, k, l\}$ and reducing by $F$ to obtain $l^\star$. This step is exact *conditioned on the triple*, but globally greedy (full $\Theta(N^4)$ exact search is infeasible at scale). Note that as of results in Table 2, we did not evaluate this for increasing the performance of the discriminator, but the results of the exact pairs is verified by the stochastic verifier.

**Complexity.** Pairs cost $\Theta(N^2)$ distance evaluations with subquadratic memory per block. Triples perform two matrix–block multiplies per $(I, J)$ tile and a per-column reduction over $k$, totaling $\Theta(N^3)$ arithmetic overall but only $O(N P_c)$ peak memory. The greedy 3$\to$4 adds a single $O(N)$ candidate sweep per retained triple.

**Verification.** We provide a GPU-side stochastic verifier: draw $S$ random $m$-plets, score them, and report (i) strict top-1 violations and (ii) exceedances above the reported $K$-th threshold. Exceedances are partitioned into those already present in the report vs. genuinely new sets; we also record worst exceedance margins. This yields a high-power consistency check without an additional exhaustive pass.

## I   ORIGINAL DATASETS $\mathrm{D}^{\mathrm{orig}}$

Table 4 summarizes key statistics of CASIA-WebFace (Yi et al., 2014), WebFace160K (Rahimi et al., 2025), and WebFace4M(Zhu et al., 2021). WebFace160K was curated to reduce the long-tail distribution of samples per identity, resulting in a more balanced dataset compared to CASIA-WebFace.

| Name | $n$ IDs | $n^r$ | Min | 25% | 50% | 75% | Max |
|---|---|---|---|---|---|---|---|
| CASIA-WebFace | ~10.5K | ~490K | 2 | 18 | 27 | 48 | 802 |
| WebFace160K | ~10K | ~160K | 11 | 13 | 16 | 19 | 24 |
| WebFace4M | ~206K | ~4,235K | 1 | 6 | 11 | 24 | 1497 |

Table 4: Summary statistics of the datasets used as $\mathrm{D}^{\mathrm{orig}}$ in this work. The middle section reports the number of identities ($n$) and real images ($n^r$). For each dataset, we also report the minimum, maximum, and 25%, 50%, and 75% percentiles of the number of samples per identity.

## J   DISCRIMINATOR DETAILS

See Tab. 5 for hardware specifications and training hyperparameters used for the IR50 and IR101 discriminators. Training on the 200K dataset will take about $2 \times 4$ 3090Ti GPU hours for the IR50 backbone and about $2.7 \times 4$ GPU hours for IR101.

Table 5: Details of the Discriminator and its Training

| Parameter Name | Discriminator T1 | Discriminator T2 | Discriminator T3 | Discriminator T4 |
|---|---|---|---|---|
| Network type | ResNet 50 | ResNet 101 | ResNet 50 | ResNet 101 |
| Marin Loss | ArcFace | ArcFace | AdaFace | AdaFace |
| Batch Size | 192 | 128 | 192 | 128 |
| GPU Number | 4 | 4 | 4 | 4 |
| Gradient Acc Step | 1 | 1 | 1 | 1 |
| GPU Type | 3090 Ti | 3090 Ti | 3090 Ti | 3090 Ti |
| FloatOpPrecision | High | High | High | High |
| MatMul Precision | High | High | High | High |
| Optimizer Type | SGD | SGD | SGD | SGD |
| Momentum | 0.9 | 0.9 | 0.9 | 0.9 |
| Weight Decay | 0.0005 | 0.0005 | 0.0005 | 0.0005 |
| Learning Rate | 0.1 | 0.1 | 0.1 | 0.1 |
| WarmUp Epoch | 1 | 1 | 1 | 1 |
| Number of Epochs | 26 | 26 | 26 | 26 |
| LR Scheduler | Step | Step | Step | Step |
| LR Milestones | [12, 24, 26] | [12, 24, 26] | [12, 24, 26] | [12, 24, 26] |
| LR Lambda | 0.1 | 0.1 | 0.1 | 0.1 |
| Input Dimension | $112 \times 112$ | $112 \times 112$ | $112 \times 112$ | $112 \times 112$ |
| Input Type | RGB images | RGB Images | RGB Images | RGB Images |
| Output Dimension | 512 | 512 | 512 | 512 |
| Seed | 2048 | 2048 | 204 8 | 2048 |

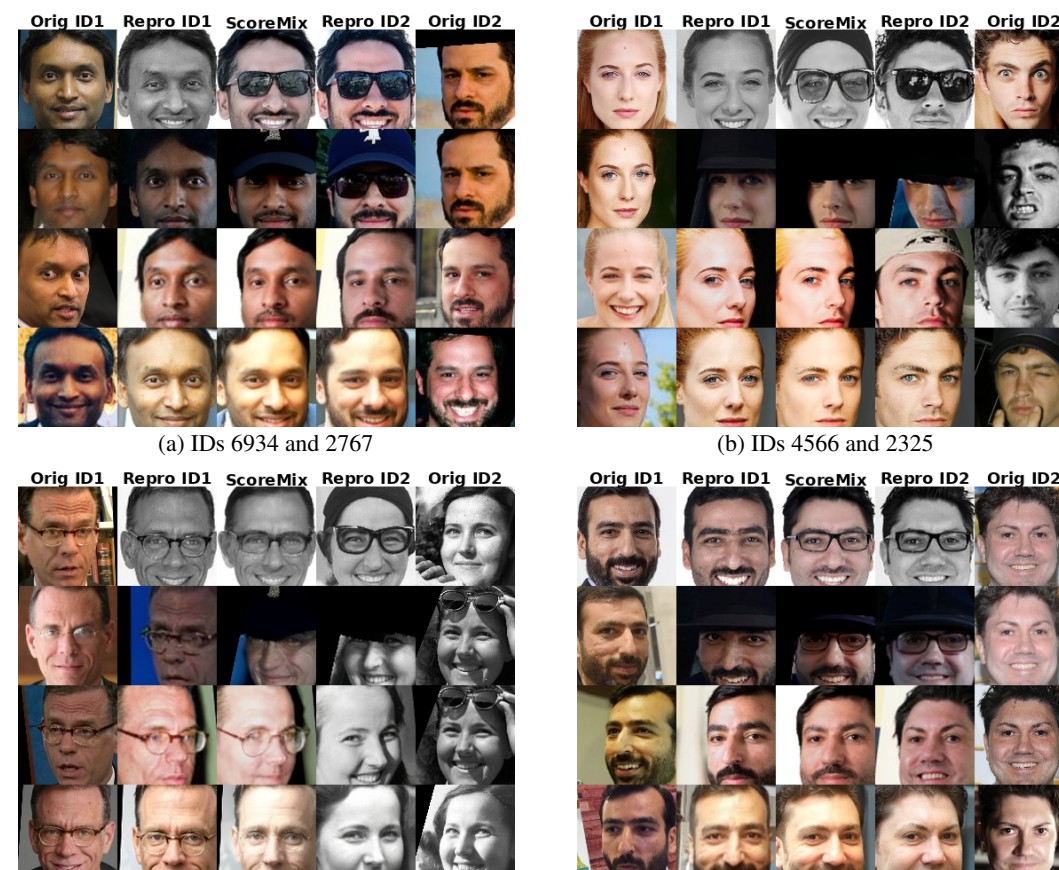

(a) IDs 6934 and 2767        (b) IDs 4566 and 2325

(c) IDs 8430 and 5412        (d) IDs 8476 and 2790

Figure 14: Qualitative comparison of ScoreMix augmentation samples. Each subfigure has five columns: from the left, *Orig ID1* and *Repro ID1* represent samples from the original dataset used to train the generator and their reproductions from the same class using the generator, respectively. Similarly, from the right, *Orig ID2* and *Repro ID2* represent samples from another identity/class. The central column (3rd from the left) shows images generated by mixing scores of ID1 and ID2 according to Equation 5 using AutoGuidance of 1.3. These images serve as augmentations for *Orig ID1* and *Orig ID2* during discriminator training. Note the subtle differences between the **ScoreMix** samples and their source counterparts; we believe these differences contribute significantly to the discriminator's improved performance beyond architectural enhancements.

## K  GENERATOR DETAILS

We used the small preset of the pixel-space EDM2 formulation, with a U-Net denoiser architecture. Training the generator required approximately 42 H100 GPU hours.

## L  MORE SAMPLES

## LLM USAGE

In accordance with the ICLR conference requirement, here we state that LLM has been used in our paper for better wording, proofreading (e.g., in long mathematical equations), and summarizing of text to better reflect the key ideas behind our work. We have also used LLMs for debugging our code and refactoring it for better readability and organization.

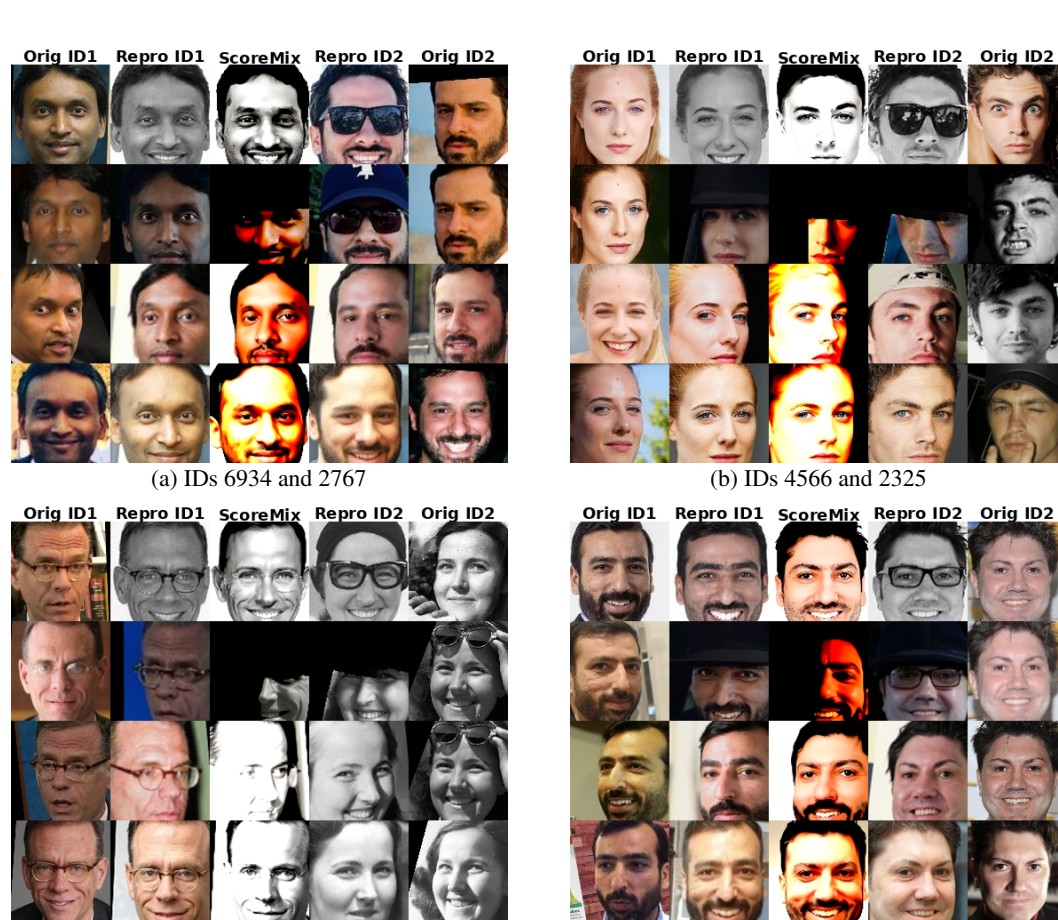

(a) IDs 6934 and 2767

(b) IDs 4566 and 2325

(c) IDs 8430 and 5412

(d) IDs 8476 and 2790

Figure 15: Qualitative comparison of ScoreMix augmentation samples. Each subfigure has five columns: from the left, *Orig ID1* and *Repro ID1* represent samples from the original dataset used to train the generator and their reproductions from the same class using the generator, respectively. Similarly, from the right, *Orig ID2* and *Repro ID2* represent samples from another identity/class. The central column (3rd from the left) shows images generated by mixing scores of ID1 and ID2 according to Equation 5 using AutoGuidance of 2.75. These images serve as augmentations for *Orig ID1* and *Orig ID2* during discriminator training. Note the subtle differences between the **ScoreMix** samples and their source counterparts; we believe these differences contribute significantly to the discriminator's improved performance beyond architectural enhancements.

IMPACT STATEMENT

In our approach, we introduce a novel technique that leverages generative models to further improve state-of-the-art (SOTA) facial recognition (FR) systems, as demonstrated on publicly available medium-sized datasets. However, these same FR systems can inadvertently facilitate unauthorized identity preservation in deepfakes and other forms of fraudulent media when attackers mimic individuals without their consent.

