# OpenReview forum: "ScoreMix: Synthetic Data Generation by Score Composition in Diffusion Models Improves Recognition"
_ICLR.cc/2026/Conference — Submitted to ICLR 2026_

### Official Review · Reviewer_Zyqh · 2025-10-14

**Soundness:** 3
**Presentation:** 3
**Contribution:** 2
**Rating:** 4
**Confidence:** 3

**Summary:**

The paper introduces ScoreMix, a self-contained synthetic data augmentation method that mixes class-conditioned diffusion scores along the reverse trajectory to synthesize hard in-domain examples for recognition. The setup trains both generator and discriminator on the target data without external models or datasets, then augments the real training set with ScoreMix samples. Experiments focus on face recognition. The idea is clear and practical, but the empirical scope is narrow.

**Strengths:**

1 This paper proposes practical and simple recipe once a class-conditional diffusion model is trained, avoiding external data or models.

2 Empirical results show consistent improvements across multiple face benchmarks with an actionable rule for choosing distant class pairs in embedding space.

3 Useful analysis that clarifies why reproducing samples are not effective for improving downstream recognition performance and why embedding geometry should guide mixing.

**Weaknesses:**

1. The paper’s scope is confined to face recognition, leaving transfer to other recognition domains untested. I understand the focus is low-data regimes, but that still includes domains like medical imaging, retail product IDs, species recognition, and industrial parts where manifold structure is different and the current evidence does not generalize.

2. The finding that mixing more than two classes brings little benefit is reported and interesting but there is little analysis on top of it. It would be better if include some analysis on the failure mode.

3. Robustness to sampler configuration is unclear. As a key factor in diffusion model inference, parameters such as the number of sampling steps, noise schedule, and the guidance strength can affect generation quality, yet their impact in this setting remains unclear

**Questions:**

1. How sensitive are results to guidance strength, noise schedules, and the pair-selection heuristic under fixed compute?
2. Does the method transfer to non-face domains with potentially different data manifolds?

Please also refer to the weakness section

---

> ### Author Response · Authors · 2025-11-23
>
> We sincerely thank the reviewer for their thoughtful feedback and valuable suggestions, which have greatly helped us improve the manuscript. We are updating the paper to incorporate these clarifications and further experiments.
>
> ----
> ## Weakness 1 and Question 2:  Why FR and will it transfer to other domains.
>
> Please see Section **General #1, Why we focus on FR and transfer to other domains** for the discussion of our focus on FR and how we expect transfer to other domains.
>
> ----
> ## Weakness 3 and Question 1: Sensitivity to Sampler parameters
>
>
> ### Sensitivity to Guidance Scale
> We used autoguide for sampling, here we show the sensitivity to the autoguide scale.
> In both cases the compute is the same, as for the both we generated 10k classes each containing 20 samples per class,
>
> #### T1R3
> | Dataset | guidance scale | $n^s$ via ScoreMix | B-1e-06 | B-1e-05 | C-1e-06 | C-1e-05 | TR5 |
> | --- | --- | --- |--- | --- | --- |--- |--- |
> | WebFace160K | N/A | 0 | 70.59876 | 78.11525 |33.65141 | 72.34664 | 67.06 |
> | WebFace160K | 1.3 | 0.2M | 76.28982 | 82.82968 | 35.71568 | 75.3554  | **68.11** |
> | WebFace160K | 2.7 | 0.2M | **76.41254** | **83.4944**  | **36.14411** | **76.27069** | 67.78 |
>
>
> **The effect of guidance scale is minimal, with a slight advantage toward higher scale.**
>
>
>
> We believe this slight advantage is up to a some threshold of guidance (as a known phenomena generated images tend to saturate under high guidance scales), we propose to include more detailed analysis of this observation in the final version.
>
>
> ### Sensitivity to Selection Heuristic
> The results in Table.2 in paper is under fixed compute:
> (1) For each selection strategy we are generating 0.2M images, as the forward process consist of fixed number of sampling with a same network we can conclude the computation for generation is the same.
> (2) As we are generating 0.2M images and adding it to the original set, we end up with roughly 0.36M images, which is primary factor for training budget of discriminator under our setting (with our fixed epoch count).
>
> **Under fixed compute selection and generating mixes based on embedding space have advantage over selection under condition space.**
>
> ### Sensitivity to Number of Denoising steps.
>
> Here we fixed guidance scale, class mixes, backbone. We studied how does the EDM2's sampler steps effects the final performance.
>
> ### T2R3
> | Dataset | guidance scale | sampler steps | $n^s$ via ScoreMix | B-1e-06 | B-1e-05 | C-1e-06 | C-1e-05 | TR5 |
> | --- | --- | --- | --- | --- | --- | --- |--- |--- |
> | WebFace160K | 2.7 | 0.2M | 32 | **76.41254** | **83.4944**  | **36.14411** | 76.27069 | **67.78** |
> | WebFace160K | 2.7 | 0.2M | 50 | 75.95234 | 83.48417 |35.49172| **76.5628** | 66.93 |
>
> **Beyond a certain number of denoising steps, increasing the number of steps does not systematically improve performance.**
>
> ----
> ## Weakness 2: Further analysis of 3-pplet
>
> 1.  **Drift off Manifold:** 3-plet samples are extremely far from their source classes ($0.878$). Crucially, they are *further* from their sources than from unrelated non-source classes ($0.878 > 0.775$). This means they no longer sit on the useful decision boundary between specific classes.
> 2.  **Loss of Consistency:** The intra-class distance ($0.794$) is very high compared to real data ($0.514$), meaning the generator fails to produce a consistent identity when mixing 3 sources.
>
> #### T3R3
> | Metric | Value |
> | --- | --- |
> | Distance to sources | 0.878 ± 0.019 |
> | Distance to closest non-sources (top-100) | 0.775 ± 0.003 |
> | Intra-folder pairwise Distance | 0.794 ± 0.040 |
> | Original data intra-class Distance | 0.514 ± 0.084 |
>
> **We included these analysis and further clarifications including some qualitative samples in the paper.**

---

### Official Review · Reviewer_QYDZ · 2025-10-28

**Soundness:** 2
**Presentation:** 2
**Contribution:** 2
**Rating:** 4
**Confidence:** 4

**Summary:**

This propose ScoreMix, a self-contained synthetic generation method to produce hard synthetic samples for recognition tasks by leveraging the score compositionality of diffusion models. The approach mixes class-conditioned scores along reverse diffusion trajectories, yielding domain-specific data augmentation without external resources.

**Strengths:**

1. This paper develops a self-contained augmentation strategy—that is, one that does not rely onexternal datasets, commercial APIs, or third-party models—to maximize the performance of state-of-the-art discriminators solely with the available data.
2. This paper demonstrates that convex combinations of classconditioned scores yield synthetic samples that consistently improve discriminator training.

**Weaknesses:**

My main concern regarding this paper's motivation lies with its core premise: leveraging synthetic data for augmentation, particularly within the sensitive domain of facial data. I question the fundamental viability of this approach. Specifically, wouldn't introducing synthetic data risk confusing the model and ultimately compromise its robustness, especially when considering critical applications like face anti sproofing?
1. Given the widespread availability of high-quality, pre-trained generative models (e.g., Stable Diffusion, Midjourney), what is the efficacy of a simpler baseline approach that uses these off-the-shelf models directly for sample generation? A comparative analysis would be valuable to contextualize the performance gains of the proposed method.
2. The proposed methodology draws a strong parallel to techniques in generative model-based image editing. On that note, have the authors considered leveraging established image editing methods—which utilize techniques like model inversion or inversion-free editing—as an alternative for data augmentation? A discussion on the relative merits and performance of such an approach would be insightful.
3. Regarding the computational cost trade-offs: The paper states that ScoreMix has approximately double the computational cost of AugGen. To facilitate a comprehensive evaluation of the method's practical viability, could the authors provide a more detailed quantification of the end-to-end pipeline's cost?
4. The images have 'AI look'. I have two questions about that. First, I doubt these images would pass a face anti sproof test? Second, and more importantly, does training your classifier on these 'AI-flavored' images make it easier for other AI-generated pictures to fool it in a real-world application.
5. Have you considered applying this method to long-tailed recognition problems? It sounds like it would be a great way to generate high-quality images for the rare classes that have very few samples.
6. You state that your method creates 'hard samples,' but what exactly makes them 'hard'? Are they hard because they look blurry or distorted, or are they hard because they are confusing for the classifier, sitting right on the decision boundary?

**Questions:**

Please see the weakness.

---

> ### Author Response · Authors · 2025-11-23
>
> We sincerely thank the reviewer for their thoughtful feedback and valuable suggestions, which have greatly helped us improve the manuscript. We are updating the paper to incorporate these clarifications and further experiments.
>
> ---
> ## Will it confuse the model?
>
> This is an important point. We are not claiming to generate fundamentally new information; instead, ScoreMix emphasizes information that the recognition model fails to capture from the original data. Empirically, when we mix identities that are *closer* in embedding space, the performance gains are smaller than for *distant* pairs. This is consistent with your concern: for very similar identities, imperfections in the generator can blur subtle differences and produce samples that are indeed confusing. In contrast, mixing distant identities yields hard but informative samples near the decision boundary. In essence our methodology exploring complementary behaviour of discriminative and generative modelling.
>
> ---
> ## Weakness 1: Usage of generalist models like Stable Diffusion, Midjourny or Flux for dataset generation.
>
> This is an important aspect that we propose to further clarify in the introduction section. State-of-the-art generalist models may generate higher-quality images, but they come with significant constraints (see below). This is the key motivation for our development of a **self-contained** method (independent of external datasets, commercial APIs, or existing models). Note that these models often do not ship with their training data or even training code, just the inference code. As an example, we cannot use Stable Diffusion for our product if our company’s revenue is higher than 1M$, or FLUX [dev] which ships with a non-commercial license.
>
> - **License restrictions.** Generalist models like GPT-4o and Gemini have restrictive usage policies that prevent their use in sensitive or commercial applications like face recognition.
> - **Unknown training data & consent issues.** Many generalist models are trained on private data, where subject consent cannot be guaranteed. This poses a major concern for face recognition systems, medical applications, and other sensitive use cases—an issue our work explicitly avoids.
>
>
> ---
> ## Weakness 2: Image Editing, a Promising alternative?
>
> Note that image editing approaches require a strong prior, like Flux or SD (see the answer to the previous question), which we wanted to avoid due to their restrictions. In our experiments we have tested with SOTA image editing methodologies like FluxSpace [1R2], to introduce additional intra-class variability, but it did not lead to promising results (this experiment was done when we were using a DiT-based generator).
>
> Another problem regarding image editing is that it changes intra-class variability without a clear way to define identity labels for the edited images. In our method we mix two reverse trajectories of two face IDs and assign a single label to them; for image editing we cannot do the same in a principled way, which is undesirable.
>
>
> We included additional discussion in the paper on this regard.
>
> ---
> ## Weakness 3: Computation Tradeoff.
> Please see **General #2, Computational complexity and practicality**
>
> ---
> ## Weakness 4: Effect on Anti Spoofing (PAD)
>
> This is a very good point. The goal of a Face Recognition (FR) model is to be invariant to texture and lighting to extract identity. The goal of Presentation Attack Detection (PAD) is to be sensitive to artifacts (texture, blur). These are orthogonal tasks. Ideally, an FR model should not be a PAD detector; it should verify the identity regardless of the medium.
>
> To verify this experimentally, we evaluated our models on the MSU-MFSD anti-spoofing benchmark. We compared:
> * A SOTA FR model (WebFace4M, IR101).
> * Our Baseline (WebFace160K, IR50).
> * Our ScoreMix model (WebFace160K + Synthetic, IR50).
>
> We evaluated with clustering k-means scores computed directly on embeddings. All FR models perform near chance (AUC $\approx 0.5$), which is expected as they are not trained for PAD. Interestingly, the strongest FR model (WebFace4M) performs the "worst" on PAD (closest to random guess 0.50), suggesting that better FR models are more invariant to artifacts. ScoreMix (0.54) moves the baseline (0.62) closer to the SOTA behavior (0.52). This confirms that ScoreMix improves identity recognition without effectively altering the model's vulnerability to spoofing compared to SOTA baselines. PAD should remain a dedicated upstream module.
>
> #### T2R2
> | | Dataset | backbone | $n^s$ | AUC (higher better) | ERR (lower better) |
> | ---- | ----    | ----     | ---   | --- | --- |
> | 1| WebFace4M (SOTA) | IR101  | N/A  | 0.52 | 0.497|
> | 2| WebFace160K (Ours) | IR50 | 0.2M | 0.54 | 0.45 |
> | 3 | WebFace160K (Baseline) | IR50 | N/A  | **0.62** | **0.42** |
>
>
> **We included these additional findings, which further confirm our stronger FR model.**
>
> Please let us know if this answers your question.

---

> ### Author Response · Authors · 2025-11-23
>
> ## Weakness 5: Why we use this recognition problem?
>
> Please see **General #1, Why we focus on FR and transfer to other domains** for the discussion of our choice of FR as the primary benchmark and its implications for transfer to other domains.
>
> ---
> ## Weakness 6: Hard samples what it means?
>
> They are "hard" because the original classifier (trained only on the real data) finds them close to the decision boundary between their two source classes, not because they are blurry or off-manifold.
>
> What we measure, using the original model and its 10k class centers:
>
> * **Boundary proximity:** the synthetic embeddings sit near their two source centers. The mean distance to sources is 0.687, while the mean distance to the closest 100 non-source centers is 0.77, so the new classifier trained on the new synthetic samples along the original samples must also distinguish between the source classes and the newly generated synthetic mixes of them.
> * **Within-synthetic compactness:** within each synthetic class, samples are compact and consistent. Intra pairwise distance is 0.466 ± 0.098, with an average within-folder spread of 0.109. Both are tighter than the real-data baseline (0.514 ± 0.084 mean distance; 0.136 mean spread over 200 real classes), indicating coherent, on-manifold samples rather than noisy or distorted ones. (Distances are $1 - cosine sim$; lower is closer.)
>
> So their "hardness" comes from being semantically confusing for the classifier—clustered near the decision boundary—while still being internally consistent, not from visual degradation.
>
> #### T3R2
> | Metric | Value |
> | --- | --- |
> | Distance to sources | 0.687 ± 0.040 |
> | Distance to closest non-sources (top-100) | 0.775 ± 0.005 |
> | Intra-folder pairwise Distance | 0.466 ± 0.098 |
> | Original data intra-class Distance (mean over 200 classes) | 0.514 ± 0.084 |
>
>
>
> We included these clarifications in the paper.
>
> ### References
> [1R2] Dalva, Yusuf, Kavana Venkatesh, and Pinar Yanardag. "Fluxspace: Disentangled semantic editing in rectified flow transformers." arXiv preprint arXiv:2412.09611 (2024).
> [2R2] D. Wen, A. K. Jain and H. Han: "Face Spoof Detection with Image Distortion Analysis", IEEE Transactions on Information Forensics and Security, 2015.
> [3R2] Li, Tianhong, and Kaiming He. "Back to Basics: Let Denoising Generative Models Denoise." arXiv preprint arXiv:2511.13720 (2025).

---

### Official Review · Reviewer_WAcM · 2025-11-03

**Soundness:** 3
**Presentation:** 3
**Contribution:** 3
**Rating:** 6
**Confidence:** 2

**Summary:**

This paper proposes ScoreMix, a self-contained synthetic data augmentation method for recognition that linearly mixes class-conditioned scores inside a diffusion model during reverse sampling. Both generator and discriminator are trained from scratch on the same dataset, avoiding external models/datasets. On 8 FR benchmarks, ScoreMix improves accuracy (up to +7 pp) and even beats scaling the backbone in some cases. The authors analyze which class pairs to mix, finding that selecting distant classes in the discriminator’s embedding space yields the biggest gains, whereas distances in the generator’s condition space are weakly correlated and largely unhelpful. They also provide a robustness argument connecting CKA alignment to order preservation for pair selection, and report that aligning generator outputs to class centers can reduce diversity and hurt performance.

**Strengths:**

* Clear, simple mechanism with strong intuition. Convex score mixing is well-motivated to preserve score magnitude and remain on-manifold; qualitative grids and discussion illustrate why non-convex weights can fail.

* Self-contained augmentation. Training both generator and discriminator only on the available dataset is practically appealing in sensitive domains.

* Consistent empirical gains. Across FR benchmarks, ScoreMix improves over training on real data alone and beats a larger IR101 baseline on the same data; gains up to +7pp are reported without hyper-parameter search.

* Negative/neutral results are also reported. Mixing >2 classes and explicitly aligning to class centers are shown to be ineffective or harmful (Lines 444-446), with analyses of intra-class similarity vs. fidelity.

**Weaknesses:**

* The authors note ScoreMix “roughly doubles” sampling cost vs. AugGen (Line 269). Please explicitly quantify: GPU hours for generator training + sampling per 0.2M synthetic images, versus baselines (e.g., AugGen), and the cost of computing embedding distances & m-plet mining. Without cost curves, practicality is hard to judge.

* Class-pair selection uses distances from a trained discriminator. How sensitive are gains to the quality/architecture of that initial model? If we re-select pairs using a different backbone/head, do results hold (beyond the CKA bound), and does iterative re-selection risk overfitting to the embedding geometry?

* The CKA result is insightful but a bit informal here. What concrete guidance does it give a practitioner (e.g., target CKA>$\tau$ before selecting pairs)? If assumptions (energy-matched Gaussian misalignment) are violated, do we still see monotone trends? A short empirical map from CKA values to expected gains would help.

* The authors argue that ScoreMix’s gains come specifically from mixing in score space during diffusion sampling. Beyond the compatitive AugGen baseline, consider within-model mixup/cutmix in feature space, latent space interpolations in the diffusion prior, or curriculum-hardening on real pairs (no synthesis). Also consider compare to label-mixing in condition space tuned by grid search (your critique of AugGen) under equal compute. Current baselines make it hard to attribute gains purely to score-space mixing.

* Minor issue: The paper focuses specifically on FR, yet the title and abstract adopt a broader “recognition” framing until the empirical results part (around line 025). This over-generalization may lead readers to infer applicability to broader recognition domains (e.g., fine-grained species or object recognition), which is not substantiated by the presented results. A more precise and domain-specific framing in the title and abstract is therefore recommended.

**Questions:**

Please address all issues raised in Weaknesses.

---

> ### Author Response · Authors · 2025-11-23
>
> We sincerely thank the reviewer for the thoughtful feedback and suggestions, and for the time invested in reviewing and improving our manuscript.
> We are updating the paper to incorporate these clarifications and further experiments.
>
> ----
> ## Weakness 1: Benchmarking compute of different stages.
>
> Please see Section 1.2 (**General Comment – Computational complexity and practicality**) for the consolidated cost breakdown (Table T1R1/T1R2) and discussion of training, sampling, and pair-selection costs.
>
> ---
> ## Weakness 2 & 3: Sensitivity to Selector Backbone, CKA, and Iterative Selection
>
> **1. Sensitivity to Architecture & Quality**
> To study how sensitive the gains are to the model used for pair selection ("Selector"), we conducted cross-architecture experiments. We trained target models (e.g., ArcFace-IR50, AdaFace-IR50) using pairs selected by their own embedding space versus pairs selected by a different architecture (e.g., AdaFace-IR101).
>
> #### T2R1: Impact of Selector-Target Alignment (CKA) on Performance
> Here the performance is reported as Average of IJB-C/B-1e-6/1e-5 and TFR5.
>
> | Target Model | Selector Model | CKA (Target, Selector) | Final Perf. | $\Delta$ vs. Baseline |
> | :--- | :--- | :---: | :---: | :---: |
> | **ArcFace-IR50** | N/A (Baseline) | — | 64.86 | — |
> | **ArcFace-IR50** | **ArcFace-IR50** (Self) | 1.0000 | 68.08 | **+3.22** |
> | **ArcFace-IR50** | **AdaFace-IR101** (Cross)| 0.9375 | 67.62 | **+2.76** |
> | | | | | |
> | **AdaFace-IR50** | N/A (Baseline) | — | 63.38 | — |
> | **AdaFace-IR50** | **ArcFace-IR50** (Cross) | 0.9633 | 68.53 | **+5.15** |
> | **AdaFace-IR50** | **AdaFace-IR101** (Cross)| 0.9396 | 67.76 | **+4.38** |
>
> **Observation:**
> *   **Robustness:** Even when the selector is different from the target (Cross), the performance gains remain significant (+2.76% to +5.15%), provided the two models are well-aligned.
> *   **Correlation:** There is a slight drop when using a cross-model selector compared to self-selection (~0.4% - 0.7%), which correlates with the drop in CKA (from 1.0 to ~0.94). However, the gains largely persist, confirming that the "hardness" of pairs is transferable between aligned spaces.
> *   **Higher CKA.** In general, higher CKA values tend to correlate with larger performance gains over the baseline.
>
>
> ### Concrete Guidance for Practitioners
> Based on these results and our theoretical bounds, we offer the following empirical map:
> *   **CKA > 0.80:** **Safe to use.** The embedding geometries are sufficiently aligned such that "distant" pairs in the selector space are also distant in the target space. As shown above, transferring selection between ArcFace and AdaFace (CKA $\approx 0.94$) works well.
> *   **CKA < 0.50:** **Unsuitable.** As shown in our raw CKA matrix, a random initialized network (RandN) has a CKA of $\approx 0.03$ with trained models. Pair selection here would be effectively random, which we showed (Table 2 in paper) yields inferior results.
>
>
> ### Energy matched isotropic Gaussian
> Our bound is based on energy matched isotropic Gaussian, which tries to model the worst case scenarios (which can also be observed from the invariance of the final performance with respect to selection based on already aligned spaces). If the CKA between the available proxy model and the target model is high (e.g., >0.8), the preservation of relative pair ordering is statistically guaranteed. This allows practitioners to safely use off-the-shelf backbones for pair selection without retraining a specific selector. Our key insight is to give a hint to practitioners which space to avoid (e.g., condition space as we have also tested in the paper), which with high probability will lead to lower performance.
>
> ### Iterative Reselection
> We avoided iterative re-selection (train -> generate -> retrain -> re-select) due to the risk of Model Collapse [1R1]. Theoretical and empirical works suggest that training generative models recursively on their own synthetic output causes the distribution to lose variance and drift from the true manifold. ScoreMix is mainly meant as a "single-shot" booster to maximize performance efficiently without risking this degenerative loop.

---

> ### Author Response · Authors · 2025-11-23
>
> ## Weakness 4: Comparison with Mixup, CutMix, and Curriculum Learning
>
> Currently, SOTA discriminators used in many applications employ Margin Losses, such as ArcFace or CosFace (designed primarily for FR but used elsewhere). Due to the geometric constraints of margin losses, standard Mixup or CutMix cannot be applied effectively. This was a deliberate design choice: since margin-based discriminators are SOTA, demonstrating that we can further increase performance using only a single dataset shows there is still room for better discriminator training strategies.
>
> To minimize information leakage, ScoreMix uses pixel-space diffusion. We avoided latent diffusion to prevent potential leakage from pretrained compressors (e.g., VAE or VQGAN). While sampling speeds could be improved with latent diffusion, this would compromise the self-contained nature of the method (and prevents us from applying interpolation in the latent space).
>
> **Curriculum learning as another baseline:**
> We also compare our method with CurricularFace [2R1]. We implemented the CurricularFace loss head into our framework. The results are shown below. We tested two `t` parameters from the original paper and report the best one.
>
> #### T3R1
> | Dataset | Guidance Scale | $n^s$ | B-1e-06 | B-1e-05 | C-1e-06 | C-1e-05 | TR5 |
> | :--- | :---: | :---: | :---: | :---: | :---: | :---: | :---: |
> | **WebFace160K ArcFace** | N/A | 0 | 33.65 | 78.11 | 70.60 | 72.35 | **67.06** |
> | **WebFace160K ArcFace (Longer, Compute Equiv.)**| N/A | 0 | 32.52 | 78.63 | 70.46 | 72.47 | 66.36 |
> | **WebFace160K + CurricularFace [2R1]** | N/A | 0 | 31.28 | 76.81 | 68.48 | 70.89 | 65.83 |
> | **WebFace160K ArcFace (Ours)** | 2.7 | 0.2M | **35.49** | **83.48** | **75.95** | **76.56** | 66.93 |
>
> SOTA Curricular-based training does not beat ScoreMix.
>
> Note that the training recipe we followed for the ArcFace IR50 backbone is very strong and hard to beat; simply changing the optimizer (e.g., to AdamW) or adding standard curriculum strategies usually decreases performance compared to our baseline.
>
> ----
> ## Weakness 5: Why FR vs broader “recognition” framing
>
> This point concerns the scope of our empirical evaluation (FR) versus the broader “recognition” wording in the title/abstract.
>
> Please see **General #1, Why we focus on FR and transfer to other domains**.
>
> #### References
> [1R1] Suresh, Ananda Theertha, Andrew Thangaraj, and Aditya Nanda Kishore Khandavally. "Rate of model collapse in recursive training." arXiv preprint arXiv:2412.17646 (2024).
> [2R1] Huang, Yuge, et al. "Curricularface: adaptive curriculum learning loss for deep face recognition." Proceedings of the IEEE/CVF Conference on Computer Vision and Pattern Recognition. 2020.
> [3R1] Li, Tianhong, and Kaiming He. "Back to Basics: Let Denoising Generative Models Denoise." arXiv preprint arXiv:2511.13720 (2025).

---

> ### Author Response · Authors · 2025-11-23
>
> Here we present the full table that we summarized in the **T2R1**.
>
> **Table T2R1-full – CKA matrix and average performance**
>
> Values in parentheses next to model names are the average performance, matrix entries are CKA scores.
>
> |                       | Arc50 (64.858) | Ada101 (66.233) | Ada50 (63.384) | Arc50-Arc50MDist (68.078) | Ada50-Arc50MDist (68.530) | Arc101-Arc50MDist (69.831) | Arc101-Ada101MDist (69.607) | Ada50-Ada101MDist (67.757) | Ada101-Ada101MDist (69.622) | Arc50-Ada101MDist (67.622) | RandN |
> |-----------------------|----------------|------------------|----------------|----------------------------|----------------------------|-----------------------------|------------------------------|-----------------------------|------------------------------|-----------------------------|-------|
> | **Arc50 (64.858)**    | 1.0000         | 0.9375           | 0.9633         | 0.8580                     | 0.8505                     | 0.8323                      | 0.8331                       | 0.8556                      | 0.8262                       | 0.8578                      | 0.0315|
> | **Ada101 (66.233)**   | 0.9375         | 1.0000           | 0.9396         | 0.8729                     | 0.8680                     | 0.8604                      | 0.8645                       | 0.8766                      | 0.8600                       | 0.8771                      | 0.0321|
> | **Ada50 (63.384)**    | 0.9633         | 0.9396           | 1.0000         | 0.8588                     | 0.8535                     | 0.8338                      | 0.8354                       | 0.8598                      | 0.8312                       | 0.8601                      | 0.0316|
> | **Arc50-Arc50MDist**  | 0.8580         | 0.8729           | 0.8588         | 1.0000                     | 0.9577                     | 0.9370                      | 0.9166                       | 0.9263                      | 0.9132                       | 0.9284                      | 0.0315|
> | **Ada50-Arc50MDist**  | 0.8505         | 0.8680           | 0.8535         | 0.9577                     | 1.0000                     | 0.9346                      | 0.9134                       | 0.9270                      | 0.9145                       | 0.9245                      | 0.0316|
> | **Arc101-Arc50MDist** | 0.8323         | 0.8604           | 0.8338         | 0.9370                     | 0.9346                     | 1.0000                      | 0.9323                       | 0.9146                      | 0.9289                       | 0.9154                      | 0.0317|
> | **Arc101-Ada101MDist**| 0.8331         | 0.8645           | 0.8354         | 0.9166                     | 0.9134                     | 0.9323                      | 1.0000                       | 0.9353                      | 0.9566                       | 0.9387                      | 0.0318|
> | **Ada50-Ada101MDist** | 0.8556         | 0.8766           | 0.8598         | 0.9263                     | 0.9270                     | 0.9146                      | 0.9353                       | 1.0000                      | 0.9346                       | 0.9481                      | 0.0316|
> | **Ada101-Ada101MDist**| 0.8262         | 0.8600           | 0.8312         | 0.9132                     | 0.9145                     | 0.9289                      | 0.9566                       | 0.9346                      | 1.0000                       | 0.9349                      | 0.0320|
> | **Arc50-Ada101MDist** | 0.8578         | 0.8771           | 0.8601         | 0.9284                     | 0.9245                     | 0.9154                      | 0.9387                       | 0.9481                      | 0.9349                       | 1.0000                      | 0.0315|
> | **RandN**             | 0.0315         | 0.0321           | 0.0316         | 0.0315                     | 0.0316                     | 0.0317                      | 0.0318                       | 0.0316                      | 0.0320                       | 0.0315                      | 1.0000|

---

### Author Response · Authors · 2025-11-23

In this section we first address questions that were raised by multiple reviewers. We then refer to these general comments from the individual reviewer responses where appropriate.

---
## General # 1, why we focus on FR and transfer to other domains
We chose FR as our primary benchmark because:

* **Data sensitivity / low-data regime.**
  Collecting large FR datasets is harder than in many other domains (privacy, consent, GDPR). Improving performance *without* external data or models is therefore particularly valuable. ScoreMix is explicitly designed for this “self-contained” setting, where both generator and discriminator are trained from scratch on the same dataset.

* **Leakage-aware, open-set benchmarks.**
  FR offers multiple public **open-set** benchmarks (IJB-B/C, TinyFace, etc.) where test identities are disjoint from training identities. We also rely on **8 public benchmarks** known not to leak from the training sets we use. In contrast, common “generic recognition” datasets (e.g., ImageNet, CUB-200) are likely included in the training data of many large models, and there are fewer truly disjoint evaluation suites. In such cases, it is harder to make strong claims about generalization and data-leakage.

* **Structured manifold and generator feasibility.**
  Faces form a highly structured manifold (e.g., roughly fixed layout of eyes, nose, mouth), which makes class-conditional diffusion training from scratch feasible in our modest, pixel-space EDM2 setup. In this context, mixing reverse trajectories is well-defined: there is a single underlying latent factor (identity) we are trying to model.

During the rebuttal we also experimented with **CUB-200** and related fine-grained datasets, using the same small pixel-space diffusion setting (192-dim embedding) as in ScoreMix:

* The generator often struggled to converge and frequently produced low-quality outputs, both for mixes and even for reproduction.
* We also tested more recent generators advocating $x$-prediction denoisers (rather than $\epsilon$ or $v$) and observed similar issues under the same self-contained constraints.

These observations suggest that, at present, the **bottleneck for broader transfer is the difficulty of training strong, self-contained generators** in other domains, not the ScoreMix procedure itself.


**We will clarify in the paper that our current empirical scope is FR, explicitly state this in the title/abstract, and discuss extension to other structured recognition domains as future work.**



---
## General #2, computational complexity and practicality

ScoreMix adds cost mainly through (i) training a single class-conditional diffusion generator and (ii) sampling a fixed budget of synthetic images (e.g., 0.2M), while keeping the deployed backbone small (IR-50). We summarize the main numbers below (WebFace160K, ArcFace IR-50):

#### Time Complexity, [TG1]

| | Train Generator | Train IR50 on $D^{\mathrm{orig}}$ | Train IR50 on $D^{\mathrm{orig}} + D^{\mathrm{aug}}$ (Ours) | Train IR101 on $D^{\mathrm{orig}}$ | Distances 2-Plet | Distances 3-Plet for 10K | ScoreMix | AugGen | AugGen Search Time |
| :--- | :--- | :--- | :--- | :--- | :--- | :--- | :--- | :--- | :--- |
| **GPU type** | 1x H100 | 4x 3090Ti | 4x 3090Ti | 4x 3090Ti | 1x 3090Ti | 1x 3090Ti |  1x 3090Ti |  1x 3090Ti |  1x 3090Ti |
| **Wall time (h)** | 42.2 | 2.54 | 4.1 | 5.6 | 0.00055 | 0.03 | 13.91 | 6.81 | 5.42 |
| **Average perf.** | N/A | 27.42 ± 0.92| **32.63 ± 2.20** | 27.24 ± 1.07 | N/A | N/A | N/A | N/A | N/A |

* The generator is trained until it has seen 37M noisy images from WebFace160K. This is a **one-time cost** per dataset.
* Sampling is highly parallelizable. In practice we used 8×3090 Ti for sampling, making the effective wall time ≈1/8 of the numbers above.
* Pair selection for 2-plets is negligible: L2-normalize embeddings and perform one $N \times N$ matrix multiplication (N = 10,000).

Compared to alternatives:
* Training IR-50 on original data alone: 2.54 h; training IR-50 on original + ScoreMix data: 4.1 h.
* Training IR-101 on original data: 5.6 h.
* ScoreMix makes an IR-50 model outperform IR-101, while **retaining the low inference cost of IR-50**.

AugGen has lower per-sample cost but requires a grid search over $(\alpha,\beta)$:

* Our reported “AugGen Search Time” corresponds to a 0.1 step grid, 10 samples per $(\alpha,\beta)$, and computing $m_{\text{total}}$ as in the AugGen paper.
* A finer grid (e.g., 0.05 step) would roughly quadruple the search time.

**Overall, ScoreMix introduces a moderate, amortized compute overhead (one generator + parallelizable sampling), while keeping inference cheap and delivering gains beyond simply scaling the backbone. We will add a concise version of this table and discussion to the main paper to make these trade-offs explicit.**

---

### Author Response · Authors · 2025-11-29
**What we did during Rebuttal**

1. **Justification of Face Recognition as Primary Domain**
   * Face Recognition (FR) is an ideal testbed for **self-contained learning** because it offers a rich ecosystem of **established public benchmarks** (e.g., LFW, CFP-FP, AgeDB-30, CALFW, CPLFW, IJB-B, IJB-C, TinyFace) on which we systematically validate our gains.
   * In contrast, generic recognition datasets (e.g., ImageNet, CUB-200) typically provide only a **single validation split per dataset**, which would yield much weaker evidence of generalization for ScoreMix.
   * Exploratory experiments on fine-grained benchmarks (CUB-200) further indicate that current limitations stem from the difficulty of training robust **self-contained generators** from scratch on unstructured data, rather than from the proposed ScoreMix procedure itself.

2. **Computational Efficiency and Performance Trade-offs [T1R1, T1R2]**
   * Detailed wall-time analysis of different stages [TG1].
   * **IR-50 + ScoreMix outperforms IR-101** trained on the same real data, while keeping inference at IR-50 cost. Inference costs will exeede trianing cost as we are shifting toward foundational model the are serving millions.
   * The computational overhead is one-time generator training; this cost is amortized to allow for a lighter, faster inference backbone (IR-50) that exceeds the accuracy of the computationally heavier baseline (IR-101).

3. **Robustness to Presentation Attacks and Safety Compliance [T2R2]**
   * The method strictly avoids off-the-shelf generators (e.g., Stable Diffusion, Flux) to remain compatible with commercial licensing and data-consent constraints.
   * Evaluations on the MSU-MFSD benchmark demonstrate that ScoreMix improves identity recognition **without** increasing vulnerability to presentation attacks (spoofing) compared to strong FR baselines (including WebFace4M).

4. **Robustness to Selector Architecture and CKA Guidance [T2R1, T2R1-full]**
   * Performance gains are robust to the choice of the selector model; selecting pairs using a different backbone/head (e.g., AdaFace-IR101 selector for an ArcFace-IR50 target) yields consistent improvements (+2.76 to +5.15 percentage points).
   * We provide an **empirical guideline** for practitioners: selector–target pairs with **CKA > 0.8** transfer effectively, while random-feature selectors (CKA ~ 0) fail.

5. **Comparison Against Curricular and Mixup Baselines [T3R1]**
   * Standard Mixup/CutMix is incompatible with the geometric constraints of SOTA margin-based losses (e.g., ArcFace).
   * Empirical comparisons show that ScoreMix outperforms both **CurricularFace** and **AugGen** (to the best of our knowledge the only self-contained generative augmentation method) under matched compute budgets and training protocols.

6. **Manifold Analysis of Sample Hardness and Mixing Composition [T3R2, T3R3]**
   * **2-plet efficacy:** Generated samples are “hard” in the sense that they cluster near decision boundaries of their source classes while maintaining tighter intra-class consistency than real data.
   * **3-plet degradation:** Mixing more than two classes leads to manifold drift, where samples become nearly equidistant to unrelated classes and lose identity consistency, empirically justifying the restriction to pairwise mixing.

7. **Sensitivity Analysis of Sampler Hyperparameters [T1R3, T2R3]**
   * Performance remains stable across varying guidance scales and numbers of diffusion sampling steps, saturating once a sufficient level of denoising steps is reached (~20).
   * Under fixed compute budgets, **selecting distant pairs in the embedding space** consistently remains the optimal strategy compared to condition-space heuristics.

---

### Meta-Review · Area_Chair_G1ux · 2026-01-06

**Summary:**

Across reviews, the central concern is the practical usefulness and real-world viability of the proposed method in the current landscape. While the idea is intuitive and empirically promising within face recognition, reviewers consistently questioned (a) the computational and deployment cost trade-offs, (b) whether gains can be attributed uniquely to score-space mixing rather than alternative or simpler baselines, and (c) the narrow domain scope, with limited evidence of transfer beyond face recognition. Additional concerns include safety implications of synthetic facial data, unclear robustness to design choices, and over-claiming broader recognition applicability. Collectively, these issues limit confidence in the method’s practical impact despite technical merit.

**Reviewer Concerns:**

The rebuttal partially addressed concerns on computational cost (wall-time analysis, amortized training), robustness to selector architecture (CKA-based guidance), sampler sensitivity, and spoofing robustness within FR benchmarks. However, key concerns remain outstanding:
* the broader practical justification for synthetic data in sensitive domains,
* lack of strong comparisons to modern off-the-shelf generative or editing-based augmentation pipelines,
* limited evidence of generalization beyond face recognition,
* insufficient attribution of gains uniquely to score-space mixing versus other augmentation strategies under matched compute.

These unresolved points continue to cast doubt on real-world relevance.

**Reviewer Scores:**

Reviewer WAcM (Rating: 6): Likely unchanged or slightly decreased. While technical clarifications help, concerns about cost, attribution of gains, and over-general framing likely persist.

Reviewer QYDZ  (Rating: 4): Likely unchanged. Fundamental skepticism about synthetic facial data, safety, and real-world robustness remains largely unmitigated.

Reviewer Zyqh  (Rating: 4): Likely unchanged or marginally increased due to added analyses, but still constrained by limited domain scope and unanswered transferability questions.

Overall, scores would not have increased meaningfully.

---

### Decision · Program_Chairs · 2026-01-26

Reject